# ASSESSING GENERALIZATION VIA DISAGREEMENT

**Yiding Jiang** [*]
Carnegie Mellon University
ydjiang@cmu.edu

**Vaishnavh Nagarajan** [*†]
Google Research
vaishnavh@google.com

**Christina Baek, J. Zico Kolter**
Carnegie Mellon University
{kbaek,zkolter}@cs.cmu.edu

## ABSTRACT

We empirically show that the test error of deep networks can be estimated by training the same architecture on the same training set but with two different runs of Stochastic Gradient Descent (SGD), and then measuring the disagreement rate between the two networks on *unlabeled test data*. This builds on — and is a stronger version of — the observation in Nakkiran & Bansal (2020), which requires the runs to be on separate training sets. We further theoretically show that this peculiar phenomenon arises from the *well-calibrated* nature of *ensembles* of SGD-trained models. This finding not only provides a simple empirical measure to directly predict the test error using unlabeled test data, but also establishes a new conceptual connection between generalization and calibration.

## 1 INTRODUCTION

Consider the following intriguing observation made in Nakkiran & Bansal (2020). Train two networks of the same architecture to zero training error on two independently drawn datasets $S_1$ and $S_2$ of the same size. Both networks would achieve a test error (or equivalently, a generalization gap) of about the same value, denoted by $\epsilon$. Now, take a fresh unlabeled dataset $U$ and measure the rate of disagreement of the predicted label between these two networks on $U$. Based on the triangle inequality, one can quickly surmise that this disagreement rate could lie anywhere between $0$ and $2\epsilon$. However, across various training set sizes and for various models like neural networks, kernel SVMs and decision trees, Nakkiran & Bansal (2020) (or N&B'20 in short) report that the disagreement rate not only linearly correlates with the test error $\epsilon$, but nearly *equals* $\epsilon$ (see first two plots in Fig 1). What brings about this unusual equality? Resolving this open question from N&B'20 could help us identify fundamental patterns in how neural networks make errors. That might further shed insight into generalization and other poorly understood empirical phenomena in deep learning.

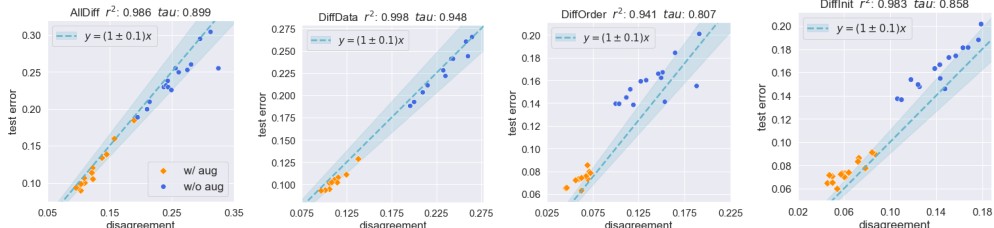

Figure 1: **GDE on CIFAR-10:** The scatter plots of pair-wise model disagreement (x-axis) vs the test error (y-axis) of the different ResNet18 trained on CIFAR10. The dashed line is the diagonal line where disagreement equals the test error. Orange dots represent models that use data augmentation. The first two plots correspond to pairs of networks trained on independent datasets, and in the last two plots, on the same dataset. The details are described in Sec 3.

In this work, we first identify a stronger observation. Consider two neural networks trained with the same hyperparameters and *the same dataset*, but with different random seeds (this could take the form e.g., of the data being presented in different random orders and/or by using a different random

---

[*]Equal contribution
[†]Work performed when Vaishnavh Nagarajan was a student at Carnegie Mellon University.

initialization of the network weights). We would expect the disagreement rate in this setting to be much smaller than in N&B'20, since both models see the same data. Yet, this is not the case: we observe on the SVHN (Netzer et al., 2011), CIFAR-10/100 (Krizhevsky et al., 2009) datasets, and for variants of Residual Networks (He et al., 2016) and Convolutional Networks (Lin et al., 2013), that the disagreement rate is still approximately equal to the test error (see last two plots in Fig 1), only slightly deviating from the behavior in N&B'20. In fact, while N&B'20 show that the disagreement rate captures significant changes in test error with varying training set sizes, we highlight a much stronger behavior: the disagreement rate is able to capture even *minute variations in the test error under varying hyperparameters like width, depth and batch size*. Furthermore, we show that under certain training conditions, these properties even hold on many kinds of *out-of-distribution* data in the PACS dataset (Li et al., 2017), albeit not on all kinds.

The above observations not only raise deeper conceptual questions about the behavior of deep networks but also crucially yield a practical benefit. In particular, our disagreement rate does not require fresh labeled data (unlike the rate in N&B'20) and rather only requires fresh *unlabeled* data. Hence, ours is a more meaningful and practical estimator of test accuracy (albeit, a slightly less accurate estimate at that). Indeed, *unsupervised accuracy estimation* is valuable for real-time evaluation of models when test labels are costly or unavailable to due to privacy considerations (Donmez et al., 2010; Jaffe et al., 2015). While there are also many other measures that correlate with generalization without access to even unlabeled test data (Jiang et al., 2018; Yak et al., 2019; Jiang et al., 2020b;a; Natekar & Sharma, 2020; Unterthiner et al., 2020), these require (a) computing intricate proportionality constants and (b) knowledge of the neural network weights/representations. Disagreement, however, provides a direct estimate, and works even with a black-box model, which makes it practically viable when the inner details of the model are unavailable due to privacy concerns.

In the second part of our work, we theoretically investigate these observations. Informally stated, we prove that if the *ensemble* learned from different stochastic runs of the training algorithm (e.g., across different random seeds) is *well-calibrated* (i.e., the predicted probabilities are neither over-confident nor under-confident), then the disagreement rate equals the test error (in expectation over the training stochasticity). Indeed, such kinds of SGD-trained deep network ensembles are known to be naturally calibrated in practice (Lakshminarayanan et al., 2017). While we do not prove *why* calibration holds in practice, the fact the condition in our theorem is empirically satisfied implies that our theory offers a valuable insight into the practical generalization properties of deep networks.

Overall, our work establishes a new connection between generalization and calibration via the idea of disagreement. This has both theoretical and practical implications in understanding generalization and the effect of stochasticity in SGD. To summarize, our contributions are as follows:

1. We prove that for any stochastic learning algorithm, *if* the algorithm leads to a well-calibrated ensemble on a particular data distribution, *then* the ensemble satisfies the *Generalization Disagreement Equality*[1] (GDE) on that distribution, *in expectation over the stochasticity of the algorithm*. Notably, our theory is general and makes no restrictions on the hypothesis class, the algorithm, the source of stochasticity, or the test distributions (which may be different from the training distribution).

2. We empirically show that for Residual Networks (He et al., 2016), convolutional neural networks (Lin et al., 2013) and fully connected networks, and on CIFAR-10/100 (Krizhevsky et al., 2009) and SVHN (Netzer et al., 2011), GDE is nearly satisfied, even on pairs of networks trained on the same data with different random seeds. This yields a simple method that in practice accurately estimates the test error using unlabeled data in these settings. We also empirically show that the corresponding ensembles are well-calibrated (according to our particular definition of calibration) in practice. We do not, however, theoretically prove why this calibration holds.

3. We present preliminary observations showing that GDE is approximately satisfied even for *certain* distribution shifts within the PACS (Li et al., 2017) dataset. This implies that the disagreement rate can be a promising estimator even for out-of-distribution accuracy.

4. We empirically find that different sources of stochasticity in SGD are almost equally effective in terms of their effect on GDE and calibration of deep models trained with SGD. We also explore the effect of pre-training on these phenomena.

---

[1]Nakkiran & Bansal (2020) refer to this as the *Agreement Property*, but we use the term Generalization Disagreement Equality to be more explicit and to avoid confusion regarding certain technical differences.

## 2 RELATED WORKS

**Understanding and predicting generalization.** Conventionally, generalization in deep learning has been studied through the lens of PAC-learning (Vapnik, 1971; Valiant, 1984). Under this framework, generalization is roughly equivalent to bounding the size of the search space of a learning algorithm. Representative works in this large area of research include Neyshabur et al. (2014; 2017; 2018); Dziugaite & Roy (2017); Bartlett et al. (2017); Nagarajan & Kolter (2019b;c); Krishnan et al. (2019). Several works have questioned whether these approaches are truly making progress toward understanding generalization in overparameterized settings (Belkin et al., 2018; Nagarajan & Kolter, 2019a; Jiang et al., 2020b; Dziugaite et al., 2020). Subsequently, recent works have proposed unconventional ways to derive generalization bounds (Negrea et al., 2020; Zhou et al., 2020; Garg et al., 2021). Indeed, even our disagreement-based estimate marks a significant departure from complexity-based approaches to generalization bounds. Of particular relevance here is Garg et al. (2021) who also leverage unlabeled data. Their bound requires modifying the original training set and then performing a careful early stopping, and is thus inapplicable to (and becomes vacuous for) interpolating models. While our estimate applies to the original training process, our guarantee applies only if we know *a priori* that the training procedure results in well-calibrated ensembles. Finally, it is worth noting that much older work (Madani et al., 2004) has provided bounds on the test error as a function of (rather than based on a "direct estimate" of) disagreement. However, these require the two runs to be on independent training sets.

While there has been research in unsupervised accuracy estimation (Donmez et al., 2010; Platanios et al., 2017; Jaffe et al., 2015; Steinhardt & Liang, 2016; ElSahar & Gallé, 2019; Schelter et al., 2020; Chuang et al., 2020), the focus has been on out-of-distribution and/or specialized learning settings. Hence, they require specialized training algorithms or extra information about the tasks. Concurrent work Chen et al. (2021) here has made similar discoveries regarding estimating accuracy via agreement, although their focus is more algorithmic than ours. We discuss this in Appendix A.

**Reducing churn.** A line of work has looked at reducing disagreement (termed there as "churn") to make predictions more reproducible and easy-to-debug. (Milani Fard et al., 2016; Jiang et al., 2021; Bhojanapalli et al., 2021). Bhojanapalli et al. (2021) further analyze how different sources of stochasticity can lead to non-trivial disagreement rates.

**Calibration.** Calibration of a statistical model is the property that the probability obtained by the model reflects the true likelihood of the ground truth (Murphy & Epstein, 1967; Dawid, 1982). A well-calibrated model provides an accurate confidence on its prediction which is paramount for high-stake decision making and interpretability. In the context of deep learning, several works (Guo et al., 2017; Lakshminarayanan et al., 2017; Fort et al., 2019; Wu & Gales, 2021; Bai et al., 2021; Mukhoti et al., 2021) have found that while individual neural networks are usually over-confident about their predictions, ensembles of several independently and stochastically trained models tend to be naturally well-calibrated. In particular, two types of ensembles have typically been studied, depending on whether the members are trained on independently sampled data also called *bagging* (Breiman, 1996) or on the same data but with different random seeds (e.g., different random initialization and data ordering) also called *deep ensembles* (Lakshminarayanan et al., 2017). The latter typically achieves better accuracy and calibration (Nixon et al., 2020).

On the theoretical side, Allen-Zhu & Li (2020) have studied why deep ensembles outperform individual models in terms of accuracy. Other works studied post-processing methods of calibration (Kumar et al., 2019), established relationships to confidence intervals (Gupta et al., 2020), and derived upper bounds on calibration error either in terms of sample complexity or in terms of the accuracy (Bai et al., 2021; Ji et al., 2021; Liu et al., 2019; Jung et al., 2020; Shabat et al., 2020).

The discussion in our paper complements the above works in multiple ways. First, most works within the machine learning literature focus on *top-class calibration*, which is concerned only with the confidence level of the top predicted class for each point. The theory in our work, however, requires looking at the confidence level of the model aggregated over all the classes. We then empirically show that SGD ensembles are well-calibrated even in this class-aggregated sense. Furthermore, we carefully investigate what sources of stochasticity result in well-calibrated ensembles. Finally, we provide an exact formal relationship between generalization and calibration via the notion of disagreement, which is fundamentally different from existing theoretical calibration bounds.

**Empirical phenomena in deep learning.** Broadly, our work falls in the area of research on identifying & understanding empirical phenomena in deep learning (Sedghi et al., 2019), especially in the context of overparameterized models that interpolate. Some example phenomena include the generalization puzzle (Zhang et al., 2017; Neyshabur et al., 2014), double descent (Belkin et al., 2019; Nakkiran et al., 2020), and simplicity bias (Kalimeris et al., 2019; Arpit et al., 2017). As stated earlier, we particularly build on N&B'20's empirical observation of the Generalization Disagreement Equality (GDE) in pairs of models trained on independently drawn datasets. They provide a proof of GDE for 1-nearest-neigbhor classifiers under specific distributional assumptions, while our result is different, and much more generic. Due to space constraints, we defer a detailed discussions of the relationship between our works in Appendix A.

## 3 DISAGREEMENT TRACKS GENERALIZATION ERROR

We demonstrate on various datasets and architectures that the test error can be estimated directly by training two runs of SGD and measuring their disagreement on an unlabeled dataset. Importantly, we show that the disagreement rate can track even minute variations in the test error induced by varying hyperparameters. Remarkably, this estimate does not require an independent labeled dataset.

**Notations.** Let $h : \mathcal{X} \to [K]$ denote a hypothesis from a hypothesis space $\mathcal{H}$, where $[K]$ denotes the set of $K$ labels $\{0, 1, \ldots, K-1\}$. Let $\mathscr{D}$ be a distribution over $\mathcal{X} \times [K]$. We will use $(X, Y)$ to denote the random variable with the distribution $\mathscr{D}$, and $(x, y)$ to denote specific values it can take. Let $\mathcal{A}$ be a stochastic training algorithm that induces a distribution $\mathscr{H}_{\mathcal{A}}$ over hypotheses in $\mathcal{H}$. Let $h, h' \sim \mathscr{H}_{\mathcal{A}}$ denote random hypotheses output by two independent runs of the training procedure. We note that the stochasticity in $\mathcal{A}$ could arise from any arbitrary source. This may arise from either the fact that each $h$ is trained on a random dataset drawn from $\mathscr{D}$ or even a completely different distribution $\mathscr{D}'$. The stochasticity could also arise from merely a different random initialization or data ordering. Next, we denote the test error and disagreement rate for hypotheses $h, h' \sim \mathscr{H}_{\mathcal{A}}$ by:

$$\text{TestErr}_{\mathscr{D}}(h) \triangleq \mathbb{E}_{\mathscr{D}} \left[ \mathbb{1}[h(X) \neq Y] \right] \quad \text{and} \quad \text{Dis}_{\mathscr{D}}(h, h') \triangleq \mathbb{E}_{\mathscr{D}} \left[ \mathbb{1}[h(X) \neq h'(X)] \right]. \quad (1)$$

Let $\tilde{h}$ denote the "ensemble" corresponding to $h \sim \mathscr{H}_{\mathcal{A}}$. In particular, define

$$\tilde{h}_k(x) \triangleq \mathbb{E}_{\mathscr{H}_{\mathcal{A}}} \left[ \mathbb{1}[h(x) = k] \right] \quad (2)$$

to be the probability value (between $[0, 1]$) given by the ensemble $\tilde{h}$ for the $k^{th}$ class. Note that the output of $\tilde{h}$ is *not* a one-hot value based on plurality vote.

**Main Experimental Setup.** We report our main observations on variants of Residual Networks, convolutional neural networks and fully connected networks trained with Momentum SGD on CIFAR-10/100, and SVHN. Each variation of the ResNet has a unique hyperparameter configuration (See Appendix C.1 for details) and all models are (near) interpolating. For each hyperparameter setting, we train two copies of models which experience two independent draws from one or more sources of stochasticity, namely 1. *random initialization* (denoted by Init) and/or 2. *ordering of a fixed training dataset* (Order) and/or 3. *different (disjoint) training data* (Data). We will use the term Diff to denote whether a source of stochasticity is "on". For example, DiffInit means that the two models have different initializations but see the same data in the same order. In DiffOrder, models share the same initialization and see the same data, but in different orders. In DiffData, the models share the initialization, but see different data. In AllDiff, the two models differ in both data and in initialization (If the two models differ in data, the training data is split into two disjoint halves to ensure no overlap.). The disagreement rate between a pair of models is computed as the proportion of the test data on which the (one-hot) predictions of the two models do not match.

**Observations.** We provide scatter plots of test error of the first run $(y)$ vs disagreement error between the two runs $(x)$ for CIFAR-10, SVHN and CIFAR-100 in Figures 1, 2 and 3 respectively (and for CNNs on CIFAR-10 in Fig 10). Naively, we would expect these scatter plots to be arbitrarily distributed anywhere between $y = 0.5x$ (if the errors of the two models are disjoint) and $x = 0$ (if the errors are identical). However, in all these scatter plots, we observe that test error and disagreement error lie very close to the diagonal line $y = x$ across different sources of stochasticity, while only slightly deviating in DiffInit/Order. In particular, in AllDiff and DiffData, the points typically

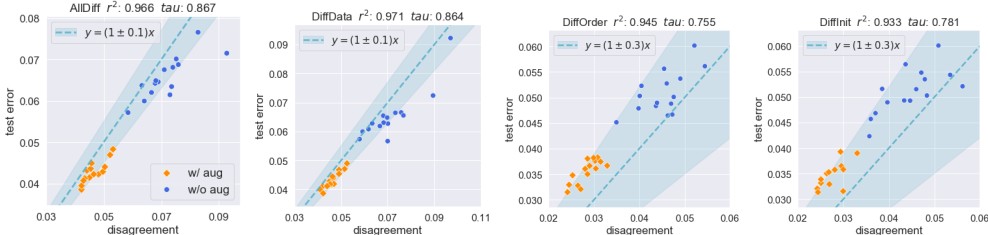

Figure 2: **GDE on SVHN:** The scatter plots of pair-wise model disagreement (x-axis) vs the test error (y-axis) of the different ResNet18 trained on SVHN.

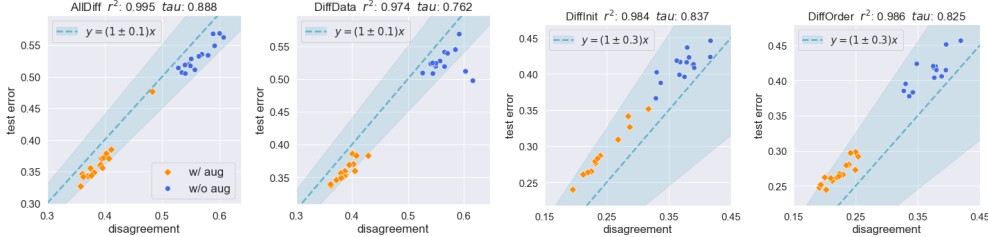

Figure 3: **GDE on CIFAR-100:** The scatter plots of pair-wise model disagreement (x-axis) vs the test error (y-axis) of the different ResNet18 trained on CIFAR100.

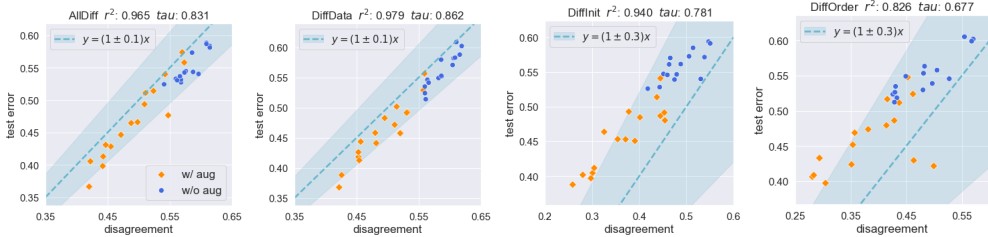

Figure 4: **GDE on 2k subset of CIFAR-10:** The scatter plots of pair-wise model disagreement (x-axis) vs the test error (y-axis) of the different ResNet18 trained on only 2000 points of CIFAR10.

lie between $y = x$ and $y = 0.9x$ while in `DiffInit` and `DiffOrder`, the disagreement rate drops slightly (since the models are trained on the same data) and so the points typically lie between $y = x$ and $y = 1.3x$. We quantify correlation via the $R^2$ coefficient and Kendall's Ranking coefficient (`tau`) reported on top of each scatter plot. Indeed, we observe that these quantities are high in all the settings, generally above $0.85$. If we focus only on the data-augmented or the non-data-augmented models, $R^2$ tends to range a bit lower, around $0.7$ and $\tau$ around $0.6$ (see Appendix D.8).

The positive observations about `DiffInit` and `DiffOrder` are surprising for two reasons. First, when the second network is trained on the *same* dataset, we would expect its predictions to be largely aligned with the original network — naturally, the disagreement rate would be negligible, and the equality observed in N&B'20 would no longer hold. Furthermore, since we calculate the disagreement rate without using a fresh labeled dataset, we would expect disagreement to be much less predictive of test error when compared to N&B'20. Our observations defy both these expectations.

There are a few more noteworthy aspects. In the low data regime where the test error is high, we would expect the models to be much less well-behaved. However, consider the CIFAR-100 plots (Fig 3), and additionally, the plots in Fig 4 where we train on CIFAR-10 with just 2000 training points. In both settings the network suffers an error as high as $0.5$ to $0.6$. Yet, we observe a behavior similar to the other settings (albeit with some deviations) — the scatter plot lies in $y = (1 \pm 0.1)x$ (for `AllDiff` and `DiffData`) and in $y = (1 \pm 0.3)x$ (for `DiffInit/Order`), and the correlation metrics are high. Similar results were established in N&B'20 for `AllDiff` and `DiffData`.

Finally, it is important to highlight that each scatter plot here corresponds to varying certain hyperparameters that cause only mild variations in the test error. Yet, the disagreement rate is able to capture those variations in the test error. This is a stronger version of the finding in N&B'20 that disagreement captures larger variations under varying dataset size.

**Effect of distribution shift and pre-training**   We study these observations in the context of the PACS dataset, a popular domain generalization benchmark with four distributions, Photo (P in short), Art (A), Cartoon (C) and Sketch (S), all sharing the same 7 classes. On any given domain, we train pairs of ResNet50 models. Both models are either randomly initialized or ImageNet (Deng et al., 2009) pre-trained. We then evaluate their test error and disagreement on all the domains. As we see in Fig 5, the surprising phenomenon here is that there *are* many pairs of source-target domains where GDE is approximately satisfied despite the distribution shift. Notably, for pre-trained models, with the exception of three pairs of source-target domains (namely, (P,C), (P,S), (S,P)), GDE is satisfied approximately. The other notable observation is that under distribution shift, pre-trained models can satisfy GDE, and often better than randomly initialized models. This is counter-intuitive, since we would expect pre-trained models to be strongly predisposed towards specific kinds of features, resulting in models that disagree rarely. See Appendix C.1 for hyperparameter details.

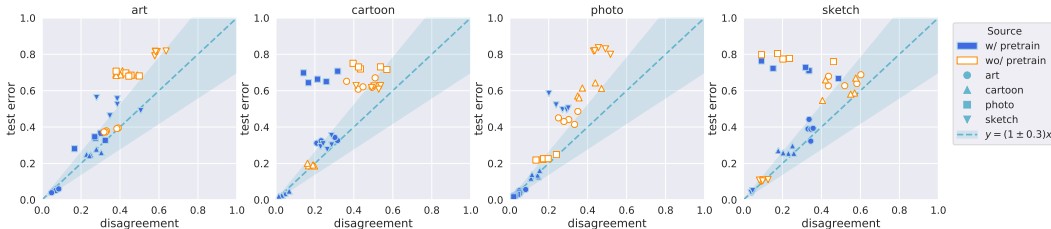

Figure 5: **GDE under distribution shift:** The scatter plots of pair-wise model disagreement (x-axis) vs the test error (y-axis) of the different ResNet50 trained on PACS. Each plot corresponds to models evaluated on the domain specified in the title. The marker shapes indicate the source domain.

## 4   CALIBRATION IMPLIES THE GDE

We now formalize our main observation, like it was formalized in N&B'20 (although with minor differences to be more general). In particular, we define "the Generalization Disagreement Equality" as the phenomenon that the test error equals the disagreement rate *in expectation over* $h \sim \mathscr{H}_\mathcal{A}$.

**Definition 4.1.** The stochastic learning algorithm $\mathcal{A}$ satisfies **the Generalization Disagreement Equality (GDE)** on the distribution $\mathscr{D}$ if,

$$\mathbb{E}_{h,h' \sim \mathscr{H}_\mathcal{A}}[\mathtt{Dis}_\mathscr{D}(h,h')] = \mathbb{E}_{h \sim \mathscr{H}_\mathcal{A}}[\mathtt{TestErr}_\mathscr{D}(h)]. \tag{3}$$

Note that the definition does not imply that the equality holds for each pair of $h, h'$ (which we observed empirically). However, for simplicity, we will stick to the above "equality in expectation" as it captures the essence of the underlying phenomenon while being easier to analyze. To motivate why proving this equality is non-trivial, let us look at the most natural hypothesis that N&B'20 identify. Imagine that all datapoints $(x, y)$ are one of two types: (a) the datapoint is so "easy" that w.p. 1 over $h \sim \mathscr{H}_\mathcal{A}, h(x) = y$ (b) the datapoint is so "hard" that $h(x)$ corresponds to picking a label uniformly at random. In such a case, with a simple calculation, one can see that the above equality would hold not just in expectation over $\mathscr{D}$, but even point-wise: for each $x$, the disagreement on $x$ in expectation over $\mathscr{H}_\mathcal{A}$ would equal the error on $x$ in expectation over $\mathscr{H}_\mathcal{A}$ (namely $(K-1)/K$ if $x$ is hard, and 0 if easy). Unfortunately, N&B'20 show that in practice, a significant fraction of the points have disagreement larger than error and another fraction have error larger than disagreement (see Appendix D.4). Surprisingly though, there is a delicate balance between these two types of points such that overall these disparities cancel each other out giving rise to the GDE.

What could create this delicate balance? We identify that this can arise from the fact that *the ensemble $\tilde{h}$ is well-calibrated*. Informally, a well-calibrated model is one whose output probability for a particular class (i.e., the model's "confidence") is indicative of the probability that the ground truth

class is indeed that class (i.e., the model's "accuracy"). There are many ways in which calibration can be formalized. Below, we provide a particular formalism called class-wise calibration.

**Definition 4.2.** The ensemble model $\tilde{h}$ satisfies **class-wise calibration** on $\mathscr{D}$ if for any confidence value $q \in [0,1]$ and for any class $k \in [K]$,

$$p(Y = k \mid \tilde{h}_k(X) = q) = q. \tag{4}$$

Next, we show that if the ensemble is class-wise calibrated on the distribution $\mathscr{D}$, then GDE does hold on $\mathscr{D}$. Note however that shortly we show a more general result where even a weaker notion of calibration is sufficient to prove GDE. But since this stronger notion of calibration is easier to understand, and the proof sketch for this captures the key intuition of the general case, we will focus on this first in detail. It is worth emphasizing that besides requiring well-calibration on the (test) distribution, all our theoretical results are general. We do not restrict the hypothesis class (it need not necessarily be neural networks), or the test/training distribution (they can be different, *as long as calibration holds in the test distribution*), or where the stochasticity comes from (it need not necessarily come from the random seed or the data).

**Theorem 4.1.** *Given a stochastic learning algorithm $\mathcal{A}$, if its corresponding ensemble $\tilde{h}$ satisfies class-wise calibration on $\mathscr{D}$, then $\mathcal{A}$ satisfies the Generalization Disagreement Equality on $\mathscr{D}$.*

*Proof.* (Sketch for binary classification. Details for full multi-class classification are deferred to App. B.2.) Let $\mathcal{X}_q$ correspond to a "confidence level set" in that $\mathcal{X}_q = \{X \in \mathcal{X} \mid \tilde{h}_0(X) = q\}$. *Our key idea is to show that for a class-wise calibrated ensemble, GDE holds within each confidence level set* i.e., for each $q \in [0,1]$, the (expected) disagreement rate equals test error for the distribution $\mathscr{D}$ restricted to the support $\mathcal{X}_q$. Since $\mathcal{X}$ is a combination of these level sets, it automatically follows that GDE holds over $\mathscr{D}$. It is worth contrasting this proof idea with the easy-hard explanation which requires showing that GDE holds point-wise, rather than confidence-level-set-wise.

Now, let us calculate the disagreement on $\mathcal{X}_q$. For any fixed $x$ in $\mathcal{X}_q$, the disagreement rate in expectation over $h, h' \sim \mathscr{H}_{\mathcal{A}}$ corresponds to $q(1-q)+(1-q)q = 2q(1-q)$. This is simply the sum of the probability of the events that $h$ predicts 0 and $h'$ predicts 1, and vice versa. Next, we calculate the test error on $\mathcal{X}_q$. At any $x$, the expected error equals $\tilde{h}_{1-y}(x)$. From calibration, we have that exactly $q$ fraction of $\mathcal{X}_q$ has the true label 0. On these points, the error rate is $\tilde{h}_1(x) = 1 - q$. On the remaining $1-q$ fraction, the true label is 1, and hence the error rate on those is $\tilde{h}_0(x) = q$. The total error rate across both the class 0 and class 1 points is therefore $q(1-q) + (1-q)q = 2q(1-q)$. $\quad\square$

**Intuition.** Even though the proof is fairly simple, it may be worth demystifying it a bit further. In short, a calibrated classifier "knows" how much error it commits in different parts of the distribution i.e., over points where it has a confidence of $q$, it commits an expected error of $1 - q$. With this in mind, it is easy to see why it is even possible to predict the test error without knowing the test labels: we can simply average the confidence values of the ensemble to estimate test accuracy. The expected disagreement error provides an alternative but less obvious route towards estimating test performance. The intuition is that within any confidence level set of a calibrated ensemble, the marginal distribution over the ground truth labels becomes identical to the marginal distribution over the one-hot predictions sampled from the ensemble. Due to this, measuring the expected disagreement of the ensemble against *itself* becomes equivalent to measuring the expected disagreement of the ensemble against *the ground truth*. The latter is nothing but the ensemble's test error.

What is still surprising though is that in practice, we are able to get away with predicting test error by computing disagreement for a single pair of ensemble members — and this works even though an ensemble of two models is *not* well-calibrated, as we will see later in Table 1. This suggests that the variance of the disagreement and test error (over the stochasticity of $\mathscr{H}_{\mathcal{A}}$) must be unusually small; indeed, we will empirically verify this in Table 1. In Corollary B.1.1, we present some preliminary discussion on why the variance could be small, leaving further exploration for future work.

### 4.1 A MORE GENERAL RESULT: CLASS-WISE TO CLASS-AGGREGATED CALIBRATION

We will now show that GDE holds under a more relaxed notion of calibration, which holds "on average" over the classes rather than individually for each class. Indeed, we demonstrate in a later section (see Appendix D.7) that this averaged notion of calibration holds more gracefully than class-wise

calibration in practice. Recall that in class-wise calibration we look at the conditional probability $p(Y = k \mid \tilde{h}_k(X) = q)$ for each $k$. Here, we will take an average of these conditional probabilities by weighting the $k^{th}$ probability by $p(\tilde{h}_k(X) = q)$. The result is the following definition:

**Definition 4.3.** The ensemble $\tilde{h}$ satisfies **class-aggregated calibration** on $\mathscr{D}$ if for each $q \in [0, 1]$,

$$\frac{\sum_{k=0}^{K-1} p(Y=k, \tilde{h}_k(X)=q)}{\sum_{k=0}^{K-1} p(\tilde{h}_k(X)=q)} = q. \tag{5}$$

**Intuition.** The denominator here corresponds to the points where *some* class gets confidence value $q$; the numerator corresponds to the points where some class gets confidence value $q$ *and* that class also happens to be the ground truth. Note however both the proportions involve counting a point $x$ multiple times if $\tilde{h}_k(x) = q$ for multiple classes $k$. In Appendix B.5, we discuss the relation between this new notion of calibration to existing definitions. In Appendix B.1 Theorem B.1, we show that the above weaker notion of calibration is sufficient to show GDE. The proof of this theorem is a non-trivial generalization of the argument in the proof sketch of Theorem 4.1, and Theorem 4.1 follows as a straightforward corollary since class-wise calibration implies class-aggregated calibration.

**Deviation from calibration.** For generality, we would like to consider ensembles that do not satisfy class-aggregated calibration precisely. How much can a deviation from calibration hurt GDE? To answer this question, we quantify calibration error as follows:

**Definition 4.4.** The **Class Aggregated Calibration Error** (CACE) of an ensemble $\tilde{h}$ on $\mathscr{D}$ is

$$\text{CACE}_{\mathscr{D}}(\tilde{h}) \triangleq \int_{q \in [0,1]} \left| \frac{\sum_k p(Y=k, \tilde{h}_k(X)=q)}{\sum_k p(\tilde{h}_k(X)=q)} - q \right| \cdot \sum_k p(\tilde{h}_k(X) = q) dq. \tag{6}$$

In other words, for each confidence value $q$, we look at the absolute difference between the left and right hand sides of Definition 4.3, and then weight the difference by the proportion of instances where a confidence value of $q$ is achieved. It is worth keeping in mind that, while the absolute difference term lies in $[0, 1]$, the weight terms alone would integrate to a value of $K$. Therefore, $\text{CACE}_{\mathscr{D}}(\tilde{h})$ can lie anywhere in the range $[0, K]$. Note that CACE is different from the "expected calibration error (ECE)" (Naeini et al., 2015; Guo et al., 2017) commonly used in the machine learning literature, which applies only to top-class calibration.

We show below that GDE holds approximately when the calibration error is low (and naturally, as a special case, holds perfectly when calibration error is zero). The proof is deferred to Appendix B.3.

**Theorem 4.2.** *For any algorithm $\mathcal{A}$, $|\mathbb{E}_{\mathscr{H}_{\mathcal{A}}}[\text{Dis}_{\mathscr{D}}(h, h')] - \mathbb{E}_{\mathscr{H}_{\mathcal{A}}}[\text{TestErr}_{\mathscr{D}}(h)]| \leq CACE_{\mathscr{D}}(\tilde{h})$.*

**Remark.** All our results hold more generally for any probabilistic classifier $\tilde{h}$. For example, if $\tilde{h}$ was an individual calibrated neural network whose predictions are given by softmax probabilities, then GDE holds for the neural network itself: the disagreement rate between two independently sampled one-hot predictions from that network would equal the test error of the softmax predictions.

## 5 Empirical Analysis of Class-aggregated Calibration

**Empirical evidence for theory.** As stated in the introduction, it is a well-established observation that ensembles of SGD trained models provide good confidence estimates (Lakshminarayanan et al., 2017). However, typically the output of these ensembles correspond to the average *softmax probabilities* of the individual models, rather than an average of the top-class predictions. Our theory is however based upon the latter type of ensembles. Furthermore, there exists many different evaluation metrics for calibration in literature, while we are particularly interested in the precise definition we have in Definition 4.3. We report our observations keeping these considerations in mind.

In Figure 6, 7 and 8, we show that SGD ensembles *do* nearly satisfy class-aggregated calibration *for all the sources of stochasticity we have considered*. In each plot, we report the conditional probability in the L.H.S of Definition 4.3 along the $y$ axis and the confidence value $q$ along the $x$ axis. We observe that the plot closely follows the $x = y$ line. In fact, we observe that calibration holds across different sources of stochasticity. We discuss this aspect in more detail in Appendix B.6.

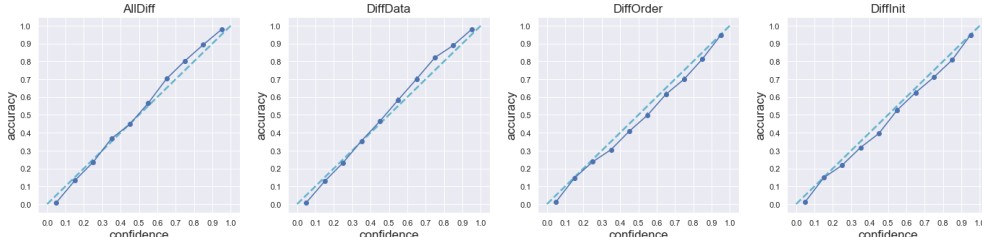

Figure 6: **Calibration on CIFAR10**: Calibration plot of different ensembles of 100 ResNet18 trained on CIFAR10. The error bar represents one bootstrapping standard deviation (most are extremely small). The estimated CACE for each scenario is shown in Table 1.

|  | **Test Error** | **Disagreement** | **Gap** | **CACE$^{(100)}$** | **CACE$^{(5)}$** | **CACE$^{(2)}$** | **ECE** |
|---|---|---|---|---|---|---|---|
| AllDiff | $0.336 \pm 0.015$ | $0.348 \pm 0.015$ | 0.012 | 0.0437 | 0.2064 | 0.4244 | 0.0197 |
| DiffData | $0.341 \pm 0.020$ | $0.354 \pm 0.020$ | 0.013 | 0.0491 | 0.2242 | 0.4411 | 0.0267 |
| DiffInit | $0.337 \pm 0.017$ | $0.307 \pm 0.022$ | 0.030 | 0.0979 | 0.2776 | 0.4495 | 0.0360 |
| DiffOrder | $0.335 \pm 0.017$ | $0.302 \pm 0.020$ | 0.033 | 0.1014 | 0.2782 | 0.4594 | 0.0410 |

Table 1: **Calibration error vs. deviation from GDE for CIFAR10:** Estimated CACE for ensembles with different number of models (denoted in the superscript) for ResNet18 on CIFAR10 with 10000 training examples. Test Error, Disagreement statistics and ECE are averaged over 100 models. Here ECE is the standard measure of top-class calibration error, provided for completeness.

For a more precise quantification of how well calibration captures GDE, we also look at our notion of calibration error, namely CACE, which also acts as an upper bound on the difference between the test error and the disagreement rate. We report CACE averaged over 100 models in Table 1 (for CIFAR-10) and Table 3 (for CIFAR-100). Most importantly, we observe that the CACE across different stochasticity settings correlates with the actual gap between the test error and the disagreement rate. In particular, CACE for AllDiff/DiffData are about 2 to 3 times smaller than that for DiffInit/Order, paralleling the behavior of $|\text{TestErr} - \text{Dis}|$ in these settings.

**Caveats.** While we believe our work provides a simple theoretical insight into how calibration leads to GDE, there are a few gaps that we do not address. First, we do not provide a theoretical characterization of *when* we can expect good calibration (and hence, when we can expect GDE). Therefore, if we do not know *a priori* that calibration holds in a particular setting, we would need labeled test data to verify if CACE is small. This would defeat the purpose of using unlabeled-data-based disagreement to measure the test error. (Thankfully, in in-distribution settings, it seems like we may be able to assume calibration for granted.) Next, our theory sheds insight into why GDE holds in expectation over training stochasticity. However, it is surprising that in practice the disagreement rate (and the test error) for a single pair of models lies close to this expectation. This occurs even though two-model-ensembles are poorly calibrated (see Tables 1 and 3). Finally, while CACE is an upper bound on the deviation from GDE, in practice CACE is only a loose bound, which could either indicate a mere lack of data/models or perhaps that our theory can be further refined.

## 6    CONCLUSION

Building on Nakkiran & Bansal (2020), we observe that remarkably, two networks trained on the same dataset, tend to disagree with each other on unlabeled data nearly as much as they disagree with the ground truth. We've also theoretically shown that this property arises from the fact that SGD ensembles are well-calibrated. Broadly, these findings contribute to the larger pursuit of identifying and understanding empirical phenomena in deep learning. Future work could shed light on why different sources of stochasticity surprisingly have a similar effect on calibration. It is also important for future work in uncertainty estimation and calibration to develop a precise and exhaustive characterization of when calibration and GDE would hold. On a different note, we hope our work inspires other novel ways to leverage unlabeled data to estimate generalization and also further cross-pollination of ideas between research in generalization and calibration.

ACKNOWLEDGMENTS

The authors would like to thank Andrej Risteski, Preetum Nakkiran and Yamini Bansal for valuable discussions during the course of this work. Yiding Jiang and Vaishnavh Nagarajan were supported by funding from the Bosch Center for Artificial Intelligence.

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

## A    RELATED WORK

Here we provide a more detailed discussion of some related work. First we discuss how our results are distinct from and/or complement their other relevant findings. First, N&B'20 provide a proof of GDE specific to 1-Nearest Neighbor models trained on two independent datasets. Our result does not restrict the hypothesis class, the algorithm or its stochasticity. Second, N&B'20 identify a notion they term as "feature calibration" which can be thought of as a generalized version of calibration. However, the instantiations of feature calibration that they empirically study are significantly different from the standard notion we study. Furthermore, they treat GDE and feature calibration as independent phenomena. Conversely, we show that calibration in the standard sense *implies* GDE. Finally, in their Appendix D.7.1, N&B'20 do report studies of ensembles where the members are trained on the same data. But this is in an altogether independent context — the GDE-related experiments in N&B'20 are all reported only on ensembles of members trained on different data. Hence, overall, their empirical results do not imply our GDE results.

In a concurrent work, Chen et al. (2021) make similar discoveries regarding estimating accuracy via agreement by developing a sophisticated ensemble-learning algorithm and make connections to calibration. Overall, their focus is more algorithmic, while our focus is primarily on understanding the nature of this phenomenon. Furthermore, in comparison to all these works which focus on out-of-distribution settings, since we focus on the in-distribution setting, we are able to identify a much simpler approach that works with *vanilla-trained blackbox* models e.g., we examine the effects of different kinds of stochasticity, we introduce ensembles only as a vehicle to understand the phenomenon theoretically (rather than an algorithmic object), and we prove how GDE holds under certain novel notions of calibration weaker than the existing ones.

## B    APPENDIX: ADDITIONAL THEORETICAL DISCUSSION

### B.1    PROOF OF THEOREM B.1

We will now prove Theorem B.1 which states that if the ensemble $\tilde{h}$ satisfies class-aggregated calibration, then the expected test error equals the expected disagreement rate.

The basic intuition behind why class-aggregated calibration implies GDE is similar to the argument for class-wise calibration — we can similarly argue that within a confidence level set, the distribution over the predicted classes matches the distribution over the ground truth labels; therefore, measuring disagreement of the ensemble against the ensemble itself boils down to measuring disagreement against the ground truth. However, the confidence level sets here can involve counting the same data point multiple times, and this nuance needs to be handled carefully as can be seen from the proof in the appendix.

**Theorem B.1.** *Given a stochastic learning algorithm $\mathcal{A}$, if its corresponding ensemble $\tilde{h}$ satisfies class-aggregated calibration (Definition 4.3) on $\mathscr{D}$, then $\mathcal{A}$ satisfies GDE on $\mathscr{D}$ (Definition 4.1).*

*Proof.*  Before we delve into the details of the proof, we will outline the high-level proof idea which is similar to that proof sketch presented in Section 4. First, we express the expected test error in terms of an integral over the confidence values (Eq 23) and then plug in the definition for class-aggregated calibration to get Eq 25. For expected disagreement rate, we can analogously express it as an integral over the confidence values, because the models are independent and the expectation naturally produces the confidence values. Note that the calibration assumption is not used for deriving the expected disagreement rate and the final result (Eq 44) is equal to the expected test error (Eq 25), which completes our proof.

We'll first simplify the expected test error and then proceed to simplifying the expected disagreement rate to the same quantity.

**Test Error**    Recall that the expected test error (which we will denote as ETE for short) corresponds to $\mathbb{E}_{\mathscr{H}_{\mathcal{A}}}\left[p(h(X) \neq Y \mid h)\right]$.

$$\mathsf{ETE} \triangleq \mathbb{E}_{h \sim \mathscr{H}_{\mathcal{A}}}\left[p(h(X) \neq Y \mid h)\right] \tag{7}$$

$$= \mathbb{E}_{h \sim \mathscr{H}_{\mathcal{A}}} \left[ \mathbb{E}_{(X,Y) \sim \mathscr{D}} \left[ \mathbb{1}[h(X) \neq Y]] \right] \right] \tag{8}$$

$$= \mathbb{E}_{(X,Y) \sim \mathscr{D}} \left[ \mathbb{E}_{\mathscr{H}_{\mathcal{A}}} \left[ \mathbb{1}[h(X) \neq Y]] \right] \right] \qquad \text{(exchanging expectations by Fubini's theorem)} \tag{9}$$

$$= \mathbb{E}_{(X,Y) \sim \mathscr{D}} \left[ 1 - \tilde{h}_Y(X) \right]. \tag{10}$$

For our further simplifications, we'll explicitly deal with integrals rather than expectations, so we get,

$$\mathsf{ETE} = \sum_{k=0}^{K-1} \int_x (1 - \tilde{h}_k(x)) p(X = x, Y = k) dx. \tag{11}$$

We'll also introduce $\tilde{h}(X)$ as a r.v. as,

$$\mathsf{ETE} = \int_{\boldsymbol{q} \in \Delta^K} \sum_{k=0}^{K-1} \int_x (1 - \tilde{h}_k(x)) p(X = x, Y = k, \tilde{h}(X) = \boldsymbol{q}) dx d\boldsymbol{q}. \tag{12}$$

Over the next few steps, we'll get rid of the integral over $x$. First, splitting the joint distribution over the three r.v.s by conditioning on the latter two,

$$\mathsf{ETE} = \int_{\boldsymbol{q} \in \Delta^K} \sum_{k=0}^{K-1} p(Y = k, \tilde{h}(X) = \boldsymbol{q}) \int_x (1 - \underbrace{\tilde{h}_k(x)}_{=q_k}) p(X = x \mid Y = k, \tilde{h}(X) = \boldsymbol{q}) dx d\boldsymbol{q} \tag{13}$$

$$= \int_{\boldsymbol{q} \in \Delta^K} \sum_{k=0}^{K-1} p(Y = k, \tilde{h}(X) = \boldsymbol{q}) \int_x \underbrace{(1 - q_k)}_{\text{constant w.r.t } \int_x} p(X = x \mid \tilde{h}(X) = \boldsymbol{q}, Y = k) dx d\boldsymbol{q} \tag{14}$$

$$= \int_{\boldsymbol{q} \in \Delta^K} \sum_{k=0}^{K-1} p(Y = k, \tilde{h}(X) = \boldsymbol{q})(1 - q_k) \underbrace{\int_x p(X = x \mid \tilde{h}(X) = \boldsymbol{q}, Y = k) dx}_{=1} d\boldsymbol{q} \tag{15}$$

$$= \underbrace{\int_{\boldsymbol{q} \in \Delta^K} \sum_{k=0}^{K-1}}_{\text{swap}} p(Y = k, \tilde{h}(X) = \boldsymbol{q})(1 - q_k) d\boldsymbol{q}. \tag{16}$$

$$= \sum_{k=0}^{K-1} \int_{\boldsymbol{q} \in \Delta^K} p(Y = k, \tilde{h}(X) = \boldsymbol{q})(1 - q_k) d\boldsymbol{q}. \tag{17}$$

$$\tag{18}$$

In the next few steps, we'll simplify the integral over $\boldsymbol{q}$ by marginalizing over all but the $k$th dimension. First, we rewrite the joint distribution of $\tilde{h}(X)$ in terms of its $K$ components. For any $k$, let $\tilde{h}_{-k}(X)$ and $\boldsymbol{q}_{-k}$ denote the $K - 1$ dimensions of both vectors excluding their $k$th dimension. Then,

$$\mathsf{ETE} = \sum_{k=0}^{K-1} \int_{q_k} \int_{\boldsymbol{q}_{-k}} p(\tilde{h}_{-k}(X) = \boldsymbol{q}_{-k} \mid Y = k, \tilde{h}_k(X) = q_k) \underbrace{p(Y = k, \tilde{h}_k(X) = q_k)(1 - q_k)}_{\text{constant w.r.t } \int_{\boldsymbol{q}_{-k}}} d\boldsymbol{q}_{-k} dq_k \tag{19}$$

$$= \sum_{k=0}^{K-1} \int_{q_k} p(Y = k, \tilde{h}_k(X) = q_k)(1 - q_k) \underbrace{\int_{\boldsymbol{q}_{-k}} p(\tilde{h}_{-k}(X) = \boldsymbol{q}_{-k} \mid Y = k, \tilde{h}_k(X) = q_k) d\boldsymbol{q}_{-k}}_{=1} dq_k \tag{20}$$

$$= \sum_{k=0}^{K-1} \int_{q_k} p(Y = k, \tilde{h}_k(X) = q_k)(1 - q_k) dq_k. \tag{21}$$

Rewriting $q_k$ as just $q$,

$$\text{ETE} = \underbrace{\sum_{k=0}^{K-1} \int_{q \in [0,1]}}_{\text{swap}} p(Y = k, \tilde{h}_k(X) = q)(1-q)dq \tag{22}$$

$$= \int_{q \in [0,1]} \sum_{k=0}^{K-1} p(Y = k, \tilde{h}_k(X) = q)(1-q)dq. \tag{23}$$

Finally, we have from the calibration in aggregate assumption that $\sum_{k=0}^{K-1} p(Y = k, \tilde{h}_k(X) = q) = q \sum_{k=0}^{K-1} p(\tilde{h}_k(X) = q)$ (Definition 4.3). So, applying this, we get

$$= \int_{q \in [0,1]} q \sum_{k=0}^{K-1} p(\tilde{h}_k(X) = q)(1-q)dq. \tag{24}$$

Rearranging,

$$\text{ETE} = \int_{q \in [0,1]} q(1-q) \sum_{k=0}^{K-1} p(\tilde{h}_k(X) = q)dq. \tag{25}$$

**Disagreement Rate** The expected disagreement rate (denoted by EDR in short) is given by the probability that two i.i.d samples from $\tilde{h}$ disagree with each other over draws of input from $\mathscr{D}$, taken in expectation over draws from $\mathscr{H}_A$. That is,

$$\text{EDR} \triangleq \mathbb{E}_{h,h' \sim \mathscr{H}_A} \left[ p(h(X) \neq h'(X) \mid h, h') \right] \tag{26}$$

$$= \mathbb{E}_{h,h' \sim \mathscr{H}_A} \left[ \mathbb{E}_{(X,Y) \sim \mathscr{D}} \left[ \mathbb{1}[h(X) \neq h'(X)] \right] \right] \tag{27}$$

$$= \mathbb{E}_{(X,Y) \sim \mathscr{D}} \left[ \mathbb{E}_{h,h' \sim \mathscr{H}_A} \left[ \mathbb{1}[h(X) \neq h'(X)] \right] \right] \quad \text{(exchanging expectations by Fubini's Theorem)} \tag{28}$$

Over the next few steps, we'll write this in terms of $\tilde{h}$ rather than $h$ and $h'$.

$$\text{EDR} = \mathbb{E}_{(X,Y) \sim \mathscr{D}} \left[ \mathbb{E}_{h,h' \sim \mathscr{H}_A} \left[ \sum_{k=0}^{K-1} \mathbb{1}[h(X) = k] \left(1 - \mathbb{1}[h'(X) = k]\right) \right] \right] \tag{29}$$

$$= \mathbb{E}_{(X,Y) \sim \mathscr{D}} \left[ \sum_{k=0}^{K-1} \mathbb{E}_{h,h' \sim \mathscr{H}_A} \left[ \mathbb{1}[h(X) = k] \left(1 - \mathbb{1}[h'(X) = k]\right) \right] \right] \quad \text{(swapping the expectation and the summation)} \tag{30}$$

$$= \mathbb{E}_{(X,Y) \sim \mathscr{D}} \left[ \sum_{k=0}^{K-1} p(h(X) = k \mid X) \left(1 - p(h'(X) = k \mid X)\right) \right] \quad \text{(since } h \text{ and } h' \text{ are i.i.d samples from } \mathscr{H}_A) \tag{31}$$

$$= \mathbb{E}_{(X,Y) \sim \mathscr{D}} \left[ \sum_{k=0}^{K-1} \tilde{h}_k(X)(1 - \tilde{h}_k(X)) \right]. \tag{32}$$

From here, we'll deal with integrals instead of expectations.

$$\text{EDR} = \int_x \sum_{k=0}^{K-1} \tilde{h}_k(x)(1 - \tilde{h}_k(x))p(X = x)dx. \tag{33}$$

Let us introduce the random variable $\tilde{h}(X)$ as,

$$\text{EDR} = \int_{\boldsymbol{q} \in \Delta^K} \int_x \sum_{k=0}^{K-1} \tilde{h}_k(x)(1 - \tilde{h}_k(x))p\left(X = x, \tilde{h}(X) = \boldsymbol{q}\right) dx d\boldsymbol{q}. \tag{34}$$

In the next few steps, we'll get rid of the integral over $x$. First, we split the joint distribution as,

$$\text{EDR} = \int_{\boldsymbol{q} \in \Delta^K} p\left(\tilde{h}(X) = \boldsymbol{q}\right) \int_x \sum_{k=0}^{K-1} \underbrace{\tilde{h}_k(x)(1 - \tilde{h}_k(x))}_{\text{apply } \tilde{h}_k(x) = q_k} p\left(X = x \mid \tilde{h}(X) = \boldsymbol{q}\right) dx d\boldsymbol{q}. \quad (35)$$

$$= \int_{\boldsymbol{q} \in \Delta^K} p\left(\tilde{h}(X) = \boldsymbol{q}\right) \int_x \underbrace{\sum_{k=0}^{K-1}}_{\text{bring to the front}} q_k(1 - q_k) p\left(X = x \mid \tilde{h}(X) = \boldsymbol{q}\right) dx d\boldsymbol{q}. \quad (36)$$

$$= \sum_{k=0}^{K-1} \int_{\boldsymbol{q} \in \Delta^K} p\left(\tilde{h}(X) = \boldsymbol{q}\right) \int_x \underbrace{q_k(1 - q_k)}_{\text{constant w.r.t } \int_x} p\left(X = x \mid \tilde{h}(X) = \boldsymbol{q}\right) dx d\boldsymbol{q}. \quad (37)$$

$$= \sum_{k=0}^{K-1} \int_{\boldsymbol{q} \in \Delta^K} p\left(\tilde{h}(X) = \boldsymbol{q}\right) q_k(1 - q_k) \underbrace{\int_x p\left(X = x \mid \tilde{h}(X) = \boldsymbol{q}\right) dx}_{1} d\boldsymbol{q}. \quad (38)$$

$$= \sum_{k=0}^{K-1} \int_{\boldsymbol{q} \in \Delta^K} p\left(\tilde{h}(X) = \boldsymbol{q}\right) q_k(1 - q_k) d\boldsymbol{q}. \quad (39)$$

Next, we'll simplify the integral over $\boldsymbol{q}$ by marginalizing over all but the $k$th dimension.

$$\text{EDR} = \sum_{k=0}^{K-1} \int_{q_k} \int_{\boldsymbol{q}_{-k}} p(\tilde{h}_{-k}(X) = \boldsymbol{q}_{-k} \mid \tilde{h}_k(X) = q_k) \underbrace{p(\tilde{h}_k(X) = q_k) q_k(1 - q_k)}_{\text{constant w.r.t. } \int_{\boldsymbol{q}_{-k}}} d\boldsymbol{q}_{-k} dq_k \quad (40)$$

$$= \sum_{k=0}^{K-1} \int_{q_k} p(\tilde{h}_k(X) = q_k) q_k(1 - q_k) \underbrace{\int_{\boldsymbol{q}_{-k}} p(\tilde{h}_{-k}(X) = \boldsymbol{q}_{-k} \mid \tilde{h}_k(X) = q_k) d\boldsymbol{q}_{-k}}_{=1} dq_k \quad (41)$$

$$= \sum_{k=0}^{K-1} \int_{q_k} p\left(\tilde{h}_k(X) = q_k\right) q_k(1 - q_k) dq_k. \quad (42)$$

Rewriting $q_k$ as just $q$,

$$\text{EDR} = \underbrace{\sum_{k=0}^{K-1} \int_{q \in [0,1]}}_{\text{swap}} p\left(\tilde{h}_k(X) = q\right) q(1 - q) dq \quad (43)$$

$$= \int_{q \in [0,1]} q(1 - q) \sum_{k=0}^{K-1} p\left(\tilde{h}_k(X) = q\right) dq. \quad (44)$$

This is indeed the same term as Eq 25, thus completing the proof.

$\qquad\qquad\qquad\qquad\qquad\qquad\qquad\qquad\qquad\qquad\qquad\qquad\qquad\qquad\qquad\qquad\qquad\square$

## B.2 PROOF OF THEOREM 4.1 AND VARIANCE OF DISAGREEMENT

Since Theorem 4.1 is a special case of Theorem B.1, we can easily prove the former:

*Proof.* Observe that if $\tilde{h}$ satisfies the class-wise calibration condition as in Definition 4.2, it must also satisfy class-aggregated calibration.

$$p\left(Y = k \mid \tilde{h}_k(X) = q\right) = q \;\; \forall k \quad (45)$$

$$\implies \frac{\sum_{i=1}^{K-1} p(Y = k, \tilde{h}_k(X) = q)}{\sum_{i=1}^{K-1} p(\tilde{h}_k(X) = q)} = \frac{\sum_{i=1}^{K-1} p\left(Y = k \mid \tilde{h}_k(X) = q\right) p(\tilde{h}_k(X) = q)}{\sum_{i=1}^{K-1} p(\tilde{h}_k(X) = q)} = q \quad (46)$$

Then, we can invoke Theorem B.1 to claim that GDE holds. $\qquad\square$

The theorem above shows that in expectation, the disagreement of independent pairs of hypotheses is equal to the test error of a single hypothesis. Now, we will drive an upper bound on the variance of the distribution. This result corroborates the empirical observation where we can use as low as a single pair of independent models to accurate estimate the test error.

**Corollary B.1.1.** *If GDE holds and there exists $\kappa \in \left[\frac{1}{2}, 1\right]$ such that, for all $h, h'$ in the support of $\mathcal{H}_\mathcal{A}$,*

$$\mathtt{Dis}(h, h') \leq \kappa \left(\mathtt{TestErr}(h) + \mathtt{TestErr}(h')\right),$$

*then*

$$Var_{h,h' \sim \mathcal{H}_\mathcal{A}} \left(\mathtt{Dis}(h, h')\right) \leq 2\kappa^2 Var_{h \sim \mathcal{H}_\mathcal{A}} \left(\mathtt{TestErr}(h)\right) + \left(4\kappa^2 - 1\right) \mathtt{ETE}^2.$$

*Proof.* First we write the expression for the exact variance of the disagreement:

$$\mathrm{Var}_{h,h' \sim \mathcal{H}_\mathcal{A}} \left(\mathtt{Dis}(h, h')\right) = \mathbb{E}_{h,h' \sim \mathcal{H}_\mathcal{A}} \left[\mathtt{Dis}(h, h')^2\right] - \mathtt{EDR}^2. \quad (47)$$

By the assumption:

$$\mathbb{E}_{h,h' \sim \mathcal{H}_\mathcal{A}} \left[\mathtt{Dis}(h, h')^2\right] \quad (48)$$

$$\leq \kappa^2 \mathbb{E}_{h,h' \sim \mathcal{H}_\mathcal{A}} \left[\left(\mathtt{TestErr}(h) + \mathtt{TestErr}(h')\right)^2\right] \quad (49)$$

$$\leq \kappa^2 \mathbb{E}_{h,h' \sim \mathcal{H}_\mathcal{A}} \left[\mathtt{TestErr}(h)^2 + 2\mathtt{TestErr}(h)\mathtt{TestErr}(h') + \mathtt{TestErr}(h')^2\right]. \quad (50)$$

Since $h$ and $h'$ are independent and identically distributed, the expectation of the cross term is equal to the product of each expectation and the second moments are equal:

$$\mathbb{E}_{h,h' \sim \mathcal{H}_\mathcal{A}} \left[\mathtt{Dis}(h, h')^2\right] \leq \kappa^2 \left(2\mathbb{E}_{h \sim \mathcal{H}_\mathcal{A}} \left[\mathtt{TestErr}(h)^2\right] + 2\mathtt{ETE}^2\right) \quad (51)$$

$$= \kappa^2 \left(2\mathbb{E}_{h \sim \mathcal{H}_\mathcal{A}} \left[\mathtt{TestErr}(h)^2\right] - 2\mathtt{ETE}^2 + 2\mathtt{ETE}^2 + 2\mathtt{ETE}^2\right) \quad (52)$$

$$= \kappa^2 \left(2\mathrm{Var}_{h \sim \mathcal{H}} \left(\mathtt{TestErr}(h)\right) + 4\mathtt{ETE}^2\right). \quad (53)$$

Substituting the inequality back to (47):

$$\mathrm{Var}_{h,h' \sim \mathcal{H}_\mathcal{A}} \left(\mathtt{Dis}(h, h')\right) \leq \kappa^2 \left(2\mathrm{Var}_{h \sim \mathcal{H}} \left(\mathtt{TestErr}(h)\right) + 4\mathtt{ETE}^2\right) - \mathtt{EDR}^2. \quad (54)$$

By the GDE, we know that $\mathtt{ETE} = \mathtt{EDR}$:

$$\mathrm{Var}_{h,h' \sim \mathcal{H}_\mathcal{A}} \left(\mathtt{Dis}(h, h')\right) \leq 2\kappa^2 \mathrm{Var}_{h \sim \mathcal{H}} \left(\mathtt{TestErr}(h)\right) + \left(4\kappa^2 - 1\right) \mathtt{ETE}^2. \quad (55)$$

$\qquad\square$

**Remark.** This corollary characterizes the worst-case behavior of the disagreement error's variance and relates it to the expectation and variance of the test error distribution over $\mathcal{H}_\mathcal{A}$. Recent works (Neal et al., 2018; Nakkiran et al., 2020) have shown that the classical bias-variance trade-off exhibits unusual behaviors in the overparameterized regime where the model contains much more parameters than the number of data points. In particular, Neal et al. (2018) shows that the variance actually decreases as the number of parameters increases, which implies that the first term of the RHS in (55) is negligible. Empirically, this is supported by the first columns of Table 1 and Table 3, where we show the variance of the test error is small.

$\kappa$ represents the amount of *structure* present in $\mathcal{H}_\mathcal{A}$. If $\kappa = 1$, the assumption $\mathtt{Dis}(h, h') \leq \mathtt{TestErr}(h) + \mathtt{TestErr}(h')$ is always true, since it follows directly from the triangle inequality; however, this would imply that there exists a pair of hypotheses in the support of $\mathcal{H}_\mathcal{A}$ that achieve the largest possible disagreement rate. Empirical evidence (Nakkiran & Bansal, 2020) indicates that this may not the case. Instead, deep models make mistakes in highly structured manner, suggesting $\kappa$ is much smaller than 1 in practice. On the other hand, if $\kappa = \frac{1}{2}$, the GDE holds pointwise for every hypotheses pair (up to the small stochasticity present in the distribution of $\mathtt{TestErr}$). This is also unlikely since it would imply the disagreement rate variance is *at most* only a half of test error variance. In practice, Table 1 and 3 show that the variance of disagreement is approximately equal to the variance of the test error, suggesting that $\kappa$ falls somewhere between $\frac{1}{2}$ and 1. This supports the hypothesis that the models make errors in a structured manner which gives rise to the observed phenomenon.

### B.3 DISAGREEMENT PROPERTY UNDER DEVIATION FROM CALIBRATION

Recall from the main paper that we quantified deviation from class-aggregated calibration in terms of CACE. Below, we provide the proof of Theorem 4.2, which shows that GDE holds approximately when CACE is low.

*Proof.* Recall from the proof of Theorem B.1 that the expected test error (ETE) satisfies:

$$\text{ETE} = \int_{q \in [0,1]} \underbrace{\sum_{k=0}^{K-1} p(Y = k, \tilde{h}_k(X) = q)}_{\text{subtract and add a } q \sum_{k=0}^{K-1} p(\tilde{h}_k(X) = q)} \quad (1 - q)dq$$

$$= \int_{q \in [0,1]} \left( \sum_{k=0}^{K-1} p(Y = k, \tilde{h}_k(X) = q) - q \sum_{k=0}^{K-1} p(\tilde{h}_k(X) = q) \right) (1 - q)dq$$

$$+ \int_{q \in [0,1]} q \sum_{k=0}^{K-1} p(\tilde{h}_k(X) = q)(1 - q)dq.$$

Recall that the second term on R.H.S is equal to the expected disagreement rate EDR. Therefore,

$$|\text{ETE} - \text{EDR}| = \int_{q \in [0,1]} \left( \sum_{k=0}^{K-1} p(Y = k, \tilde{h}_k(X) = q) - q \sum_{k=0}^{K-1} p(\tilde{h}_k(X) = q) \right) (1 - q)dq.$$

Multiplying and dividing the inner term by $\sum_{k=0}^{K-1} p(\tilde{h}_k(X) = q)$,

$$|\text{ETE} - \text{EDR}| = \left| \int_{q \in [0,1]} \left( \frac{\sum_{k=0}^{K-1} p(Y = k, \tilde{h}_k(X) = q)}{\sum_{k=0}^{K-1} p(\tilde{h}_k(X) = q)} - q \right) \sum_{k=0}^{K-1} p(\tilde{h}_k(X) = q)(1 - q)dq \right|$$

$$\leq \int_{q \in [0,1]} \left| \frac{\sum_{k=0}^{K-1} p(Y = k, \tilde{h}_k(X) = q)}{\sum_{k=0}^{K-1} p(\tilde{h}_k(X) = q)} - q \right| \sum_{k=0}^{K-1} p(\tilde{h}_k(X) = q) \underbrace{(1 - q)}_{\leq 1} dq$$

$$\leq \int_{q \in [0,1]} \left| \frac{\sum_{k=0}^{K-1} p(Y = k, \tilde{h}_k(X) = q)}{\sum_{k=0}^{K-1} p(\tilde{h}_k(X) = q)} - q \right| \sum_{k=0}^{K-1} p(\tilde{h}_k(X) = q)dq$$

$$= \text{CACE}(\tilde{h}).$$

$\square$

Note that it is possible to consider a more refined definition of CACE that yields a tighter bound on the gap. In particular, in the last series of equations, we can leave the $1 - q$ as it is, without upper bounding by $1$. In practice, this tightens CACE by upto a value of $2$. We however avoid considering the refined definition as it is less intuitive as an error metric.

### B.4 CALIBRATION IS NOT A NECESSARY CONDITION FOR GDE

Theorem B.1 shows that calibration implies GDE. Below, we show that the converse is not true. That is, if the ensemble satisfies GDE, it is not necessarily the case that it satisfies class-aggregated calibration. This means that calibration and GDE are not equivalent phenomena, but rather only that calibration may lead to the latter.

**Proposition B.1.** *For a stochastic algorithm $\mathcal{A}$ to satisfy GDE, it is not necessary that its corresponding ensemble $\tilde{h}$ satisfies class-aggregated calibration.*

*Proof.* Consider an example where $\tilde{h}$ assigns a probability of either $0.1$ or $0.2$ to class $0$. In particular, assume that with $0.5$ probability over the draws of $(x, y) \sim \mathcal{D}$, $\tilde{h}_0(x) = 0.1$ and with

0.5 probability, $\tilde{h}_0(x) = 0.2$. The expected disagreement rate (EDR) of this classifier is given by $\mathbb{E}_{\mathscr{D}}\left[2\tilde{h}_0(x)\tilde{h}_1(x)\right] = 2\left(\frac{0.1 \cdot 0.9 + 0.2 \cdot 0.8}{2}\right) = 0.25$.

Now, it can be verified that the binary classification setting, class-aggregated and class-wise calibration are identical. Therefore, letting $p(Y = 0 \mid \tilde{h}_0(X) = 0.1) \triangleq \epsilon_1$ and $p(Y = 0 \mid \tilde{h}_0(X) = 0.2) \triangleq \epsilon_2$, our goal is to show that it is possible for $\epsilon_1 \neq 0.1$ or $\epsilon_2 \neq 0.2$ and still have the expected test error (ETE) equal the EDR of 0.25. Now, the ETE on $\mathscr{D}$ conditioned on $\tilde{h}_0(x) = 0.1$ is given by $(0.1(1 - \epsilon_1) + 0.9\epsilon_1)$ and on $\tilde{h}_0(x) = 0.2$ is given by $(0.2(1 - \epsilon_2) + 0.8\epsilon_1)$. Thus, the ETE on $\mathscr{D}$ is given by $0.15 + 0.5(0.8\epsilon_1 + 0.6\epsilon_2)$. We want $0.15 + 0.5(0.8\epsilon_1 + 0.6\epsilon_2) = 0.25$ or in other words, $0.8\epsilon_1 + 0.6\epsilon_2 = 0.2$.

Observe that while $\epsilon_1 = 0.1$ and $\epsilon_2 = 0.2$ is one possible solution where $\tilde{h}$ would satisfy class-wise calibration/class-aggregated calibration, there are also infinitely many other solutions for this equality to hold (such as say $\epsilon_1 = 0.25$ and $\epsilon_2 = 0$) where calibration does not hold. Thus, class-aggregated/class-wise calibration is just one out of infinitely many possible ways in which $\tilde{h}$ could be configured to satisfy GDE. $\qquad\square$

### B.5 COMPARING CACE TO EXISTING NOTIONS OF CALIBRATION.

Calibration in machine learning literature (Guo et al., 2017; Nixon et al., 2019) is often concerned only with the confidence level of the top predicted class for each point. While top-class calibration is weaker than class-wise calibration, it is neither stronger nor weaker than class-aggregated calibration. Class-wise calibration is a notion of calibration that has appeared originally under different names in Zadrozny & Elkan; Wu & Gales (2021). On the other hand, the only closest existing notion to class-aggregated calibration seems to be that of *static calibration* in Nixon et al. (2019), where it is only indirectly defined. Another existing notion of calibration for the multi-class setting is that of *strong calibration* (Vaicenavicius et al., 2019; Widmann et al., 2019) which evaluates the accuracy of the model conditioned on $\tilde{h}(X)$ taking a particular value in the $K$-simplex. This is significantly stronger than class-wise calibration since this would require about $\exp(K)$ many equalities to hold rather than just the $K$ equalities in Definition 4.2.

### B.6 THE EFFECT OF DIFFERENT STOCHASTICITY.

Compared to `AllDiff/DiffData`, `DiffInit/Order` is still well-calibrated, with only slight deviations. Why is varying the training data almost as effective in calibration as varying the random seed? One might propose the following natural hypothesis in the context of `DiffOrder` vs `DiffData`. In the first few steps of SGD, the data seen under two different reorderings are likely to not intersect at all, and hence the two trajectories would *initially* behave as though being trained on two independent datasets. Further, if the first few steps largely determine the kind of minimum that the training falls into, then it is reasonable to expect that the stochasticity in data and in ordering both have the same effect on calibration. However, this hypothesis falls apart when we try to understand why two runs with the *same ordering* and *different initialization* (`DiffInit`) exhibits the same effect as `DiffData`. Indeed, Fort et al. (2019) have empirically shown that two such SGD runs explore diverse regions in the function space. Hence, we believe that there is a more nuanced reason behind why different types of stochasticity have a similar effect on ensemble calibration. One promising hypothesis for this could be the multi-view hypothesis from Allen-Zhu & Li (2020), which show that different random initializations could encourage the network to latch on to different predictive features (even when exposed to the same training set), and thus result in ensembles with better test accuracy. Extending their study to understand similar effects on calibration would be a useful direction for future research.

## C  APPENDIX: EXPERIMENTAL DETAILS

### C.1  PAIRWISE DISAGREEMENT

In this work, the main architectures we used are ResNet18 with the following hyperparameter configurations:

1. width multiplier: $\{1\times, 2\times\}$
2. initial learning rate: $\{0.1, 0.05\}$
3. weight decay: $\{0.0001, 0.0\}$
4. minibatch size: $\{200, 100\}$
5. data augmentation: $\{No, Yes\}$

Width multiplier refers to how much wider the model is than the architecture presented in He et al. (2016) (i.e. every filter width is multiplied by the width multiplier). All models are trained with SGD with momentum of 0.9. The learning rate decays $10\times$ every 50 epochs. The training stops when the training accuracy reaches 100%.

For Convolutional Neural Network experiments, we use architectures similar to Network-in-Network (Lin et al., 2013). On a high level, the architecture contains blocks of $3 \times 3$ convolution followed by two $1 \times 1$ convolution (3 layers in total). Each block has the same width and the final layer is projected to output class number with another $1 \times 1$ convolution followed by a global average pooling layer to yield the final logits. Other differences from the original implementation are that we do not use dropout and add batch normalization layer is added after every layer. The hyperparameters are:

1. depth: $\{7, 10, 13\}$
2. width: $\{128, 256, 384\}$
3. weight decay: $\{0.001, 0.0\}$
4. minibatch size: $\{200, 100, 300\}$

All models are optimized with momentum of 0.9 and uses the same learning rate schedule as ResNet18.

For Fully Connected Networks, we use:

1. depth: $\{1,2,3,4\}$
2. width: $\{128, 256, 384, 512\}$
3. weight decay: $\{0.0, 0.001\}$
4. minibatch size: $\{100, 200, 300\}$

All models are optimized with momentum of 0.9 and uses the same learning rate schedule as ResNet18.

**Distribution shift and pre-training experiments.** On all our experiments on the PACS dataset, we use ResNet50 (with Batch Normalization layers frozen and the final fully-connected layer removed) as our featurizer and one linear layer as our classifier. All our models are trained until 3000 steps after reaching 0.995 training accuracy with the following hyperparameter configurations:

1. learning rate: 0.00005
2. weight decay: 0.0
3. learning rate decay: None
4. minibatch size: 100
5. data augmentation: Yes

On each of these domains, we train 5 pairs of ResNet50 with a linear layer on top, varying the random seeds (keeping hyperparameters constant). Both models in a pair are trained on the same $80\%$ of the data, and only differ in their initialization, data ordering and the augmentation on the data. We then evaluate the test error and disagreement rate of all pairs on each of the four domains. We consider both randomly initialized models and ImageNet pre-trained models (Deng et al., 2009). For pre-trained models, only the linear layer is initialized differently between the two models in a pair.

## C.2 ENSEMBLE

For ensembles, unless specified otherwise, we use the combination of the first option for each hyperparameters in Section C.1.

### C.2.1 FINITE-SAMPLE APPROXIMATION OF CACE

For every ensemble experiment, we train a standard ResNet 18 model (width multiplier $1\times$, initial learning rate 0.1, weight decay 0.0001, minibatch size 200 and no data augmentation).

To estimate the calibration, we use the testset $\mathcal{D}_{test}$. We split $[0, 1]$ into 10 equally sized bins. For a class $k$, we can group all $(x, y) \in \mathcal{D}_{test}$ into different bins $\mathcal{B}_i^k$ according to $\tilde{h}_k(x)$ (all bins have boundaries that do not overlap with other bins). In total, there are $10 \times K$ bins.

$$\mathcal{B}_i^k = \left\{ (x, y) \mid \text{lower}(\mathcal{B}_i^k) \leq \tilde{h}_k(x) < \text{upper}(\mathcal{B}_i^k) \text{ and } (x, y) \in \mathcal{D}_{test} \right\} \tag{56}$$

Where *upper* and *lower* are the boundaries of the bin. To mitigate the effect of insufficient samples for some of the middling confidence value in the middle (e.g. $p = 0.5$), we further aggregate the calibration accuracy over the classes into a single bin $\mathcal{B}_i = \bigcup_{k=1}^{K} \mathcal{B}_i^k$ in a weighted manner. Concretely, for each bin, we sum over all the classes when computing the accuracy:

$$\text{acc}(\mathcal{B}_i) = \frac{1}{\sum_{k=1}^{K} |\mathcal{B}_i^k|} \sum_{k=1}^{K} \sum_{(x,y) \in \mathcal{B}_i^k} \mathbb{1}[y = k] = \frac{1}{|\mathcal{B}_i|} \sum_{k=1}^{K} \sum_{(x,y) \in \mathcal{B}_i^k} \mathbb{1}[y = k] \tag{57}$$

To quantify the how "far" the ensemble is from the ideal calibration level, we use the Class Aggregated Calibration Error (CACE) which is an average of how much each bin deviates from $y = x$ weighted by the number of samples in the bin:

$$\widehat{CACE} = \sum_{i=1}^{N_{\mathcal{B}}} \frac{|\mathcal{B}_i|}{|\mathcal{D}_{test}|} \left| \text{acc}(\mathcal{B}_i) - \text{conf}(\mathcal{B}_i) \right| \tag{58}$$

where $N_{\mathcal{B}}$ is number of bins (usually 10 in this paper unless specified otherwise), $\text{conf}(\mathcal{B}_i)$ is the ideal confidence level of the bin, which we set to the average confidence of all data points in the bin. This is the sample-based approximation of definition 4.4.

### C.2.2 FINITE-SAMPLE APPROXIMATION OF ECE

ECE is a widely used metric for measuring calibration. For completeness, we will reproduce its approximation here. Let $\hat{Y}$ be the class with highest probability under $\tilde{h}$ (we are omitting the dependency on $X$ in the notation since it is clear):

$$\hat{Y} = \arg\max_{k \in [K]} \tilde{h}_k(X) \tag{59}$$

We once again split $[0, 1]$ into 10 equally sized bins but do not divide further into $K$ classes. Each bin is constructed as:

$$\mathcal{B}_i = \left\{ (x, y) \mid \text{lower}(\mathcal{B}_i) \leq \tilde{h}_{\hat{y}}(x) < \text{upper}(\mathcal{B}_i) \text{ and } (x, y) \in \mathcal{D}_{test} \right\} \tag{60}$$

With the same notation used for CACE, the accuracy is computed as:

$$\text{acc}(\mathcal{B}_i) = \frac{1}{|\mathcal{B}_i|} \sum_{(x,y) \in \mathcal{B}_i} \mathbb{1}[y = \hat{y}] \tag{61}$$

Finally, the approximation of ECE is computed as the following:

$$\widehat{ECE} = \sum_{i=1}^{N_{\mathcal{B}}} \frac{|\mathcal{B}_i|}{|\mathcal{D}_{test}|} \, |\text{acc}(\mathcal{B}_i) - \text{conf}(\mathcal{B}_i)| \tag{62}$$

# D    ADDITIONAL EMPIRICAL RESULTS

## D.1    ADDITIONAL FIGURES

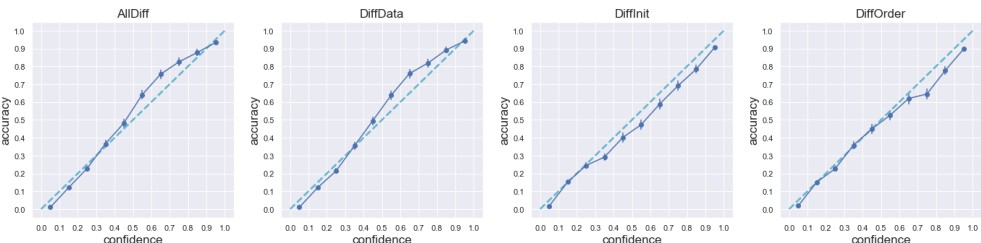

Figure 7: **Calibration on 2k subset of CIFAR10**: Calibration plot of different ensembles of 100 ResNet18 trained on CIFAR10 with 2000 training points.

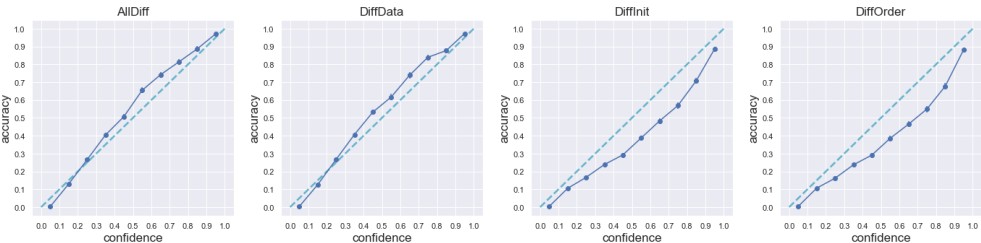

Figure 8: **Calibration on CIFAR100**: Calibration plot of different ensembles of 100 ResNet18 trained on CIFAR100 with 10000 data points.

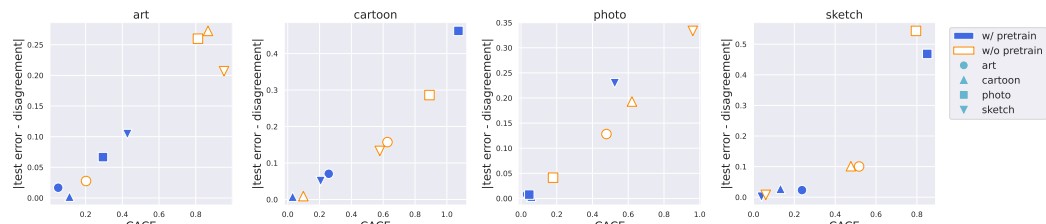

Figure 9: **Calibration error vs. deviation from GDE under distribution shift**: The scatter plots of CACE (x-axis) vs the gap between the test error and disagreement rate (y-axis) averaged over an ensemble of 10 ResNet50 models trained on PACS. Each plot corresponds to models evaluated on the domain specified in the title. The source/training domain is indicated by different marker shapes.

## D.2    CIFAR100 CALIBRATION TABLE

Here we present the calibration error for ResNet18 trained on CIFAR100 with 10k training examples.

## D.3    OTHER DATASETS AND ARCHITECTURES

In Fig 10, we provide scatter plots for fully-connected networks (FCN) on MNIST, and convolutional networks (CNN) on CIFAR10. We observe that when trained on the whole MNIST dataset, there is larger deviation from the $x = y$ behavior (see left-most image). But when we reduce the dataset size to 2000, we recover the GDE observation on MNIST. We observe that the CNN settings satisfies GDE too.

|  |  | **Test Error** | **Disagreement** | **Gap** | **CACE**[(10)] |
|---|---|---|---|---|---|
| | Art $\rightarrow$ Art | $0.0518 \pm 0.0107$ | $0.0685 \pm 0.0133$ | 0.0166 | 0.0505 |
| | Art $\rightarrow$ Cartoon | $0.3229 \pm 0.0365$ | $0.2524 \pm 0.0446$ | 0.0705 | 0.2577 |
| | Art $\rightarrow$ Photo | $0.0509 \pm 0.0097$ | $0.0592 \pm 0.0136$ | 0.0082 | 0.0335 |
| | Art $\rightarrow$ Sketch | $0.3871 \pm 0.0613$ | $0.3639 \pm 0.079$ | 0.0231 | 0.2374 |
| | Cartoon $\rightarrow$ Art | $0.2555 \pm 0.0203$ | $0.2534 \pm 0.0262$ | 0.0020 | 0.1121 |
| | Cartoon $\rightarrow$ Cartoon | $0.0303 \pm 0.0118$ | $0.0380 \pm 0.0150$ | 0.0077 | 0.0308 |
| | Cartoon $\rightarrow$ Photo | $0.1361 \pm 0.0227$ | $0.1327 \pm 0.0201$ | 0.0034 | 0.0580 |
| Pretrained | Cartoon $\rightarrow$ Sketch | $0.2672 \pm 0.0201$ | $0.2398 \pm 0.0326$ | 0.0273 | 0.1322 |
| | Photo $\rightarrow$ Art | $0.3315 \pm 0.0487$ | $0.2649 \pm 0.0624$ | 0.0666 | 0.2943 |
| | Photo $\rightarrow$ Cartoon | $0.6721 \pm 0.0545$ | $0.2100 \pm 0.0601$ | 0.4621 | 1.0725 |
| | Photo $\rightarrow$ Photo | $0.0245 \pm 0.0110$ | $0.0322 \pm 0.0125$ | 0.0076 | 0.0460 |
| | Photo $\rightarrow$ Sketch | $0.7180 \pm 0.0774$ | $0.2497 \pm 0.1350$ | 0.4683 | 0.8507 |
| | Sketch $\rightarrow$ Art | $0.5187 \pm 0.0598$ | $0.4144 \pm 0.0769$ | 0.1042 | 0.4274 |
| | Sketch $\rightarrow$ Cartoon | $0.3064 \pm 0.0330$ | $0.2557 \pm 0.0349$ | 0.0506 | 0.2069 |
| | Sketch $\rightarrow$ Photo | $0.5203 \pm 0.0435$ | $0.2908 \pm 0.0618$ | 0.2295 | 0.5245 |
| | Sketch $\rightarrow$ Sketch | $0.0443 \pm 0.0067$ | $0.0460 \pm 0.0074$ | 0.0016 | 0.0389 |

|  |  | **Test Error** | **Disagreement** | **Gap** | **CACE**[(10)] |
|---|---|---|---|---|---|
| | Art $\rightarrow$ Art | $0.3821 \pm 0.0137$ | $0.3545 \pm 0.0361$ | 0.0276 | 0.2021 |
| | Art $\rightarrow$ Cartoon | $0.6337 \pm 0.0320$ | $0.4764 \pm 0.0481$ | 0.1573 | 0.6267 |
| | Art $\rightarrow$ Photo | $0.4443 \pm 0.0277$ | $0.3161 \pm 0.0360$ | 0.12821 | 0.4778 |
| | Art $\rightarrow$ Sketch | $0.6517 \pm 0.0337$ | $0.5513 \pm 0.0697$ | 0.1004 | 0.5169 |
| | Cartoon $\rightarrow$ Art | $0.6911 \pm 0.0158$ | $0.4186 \pm 0.0526$ | 0.2725 | 0.8669 |
| | Cartoon $\rightarrow$ Cartoon | $0.1910 \pm 0.0163$ | $0.1817 \pm 0.0221$ | 0.0092 | 0.0981 |
| | Cartoon $\rightarrow$ Photo | $0.6 \pm 0.0439$ | $0.4070 \pm 0.0522$ | 0.1929 | 0.6207 |
| Not Pretrained | Cartoon $\rightarrow$ Sketch | $0.6084 \pm 0.0720$ | $0.5072 \pm 0.0757$ | 0.1012 | 0.4767 |
| | Photo $\rightarrow$ Art | $0.6899 \pm 0.0149$ | $0.4301 \pm 0.0385$ | 0.8119 | 0.2598 |
| | Photo $\rightarrow$ Cartoon | $0.7293 \pm 0.0222$ | $0.4431 \pm 0.0566$ | 0.8905 | 0.2862 |
| | Photo $\rightarrow$ Photo | $0.2281 \pm 0.0168$ | $0.1868 \pm 0.0258$ | 0.1791 | 0.0413 |
| | Photo $\rightarrow$ Sketch | $0.7823 \pm 0.021$ | $0.2388 \pm 0.1180$ | 0.7954 | 0.5435 |
| | Sketch $\rightarrow$ Art | $0.8110 \pm 0.0164$ | $0.6042 \pm 0.0629$ | 0.2068 | 0.9540 |
| | Sketch $\rightarrow$ Cartoon | $0.6215 \pm 0.0114$ | $0.4882 \pm 0.0522$ | 0.1332 | 0.5785 |
| | Sketch $\rightarrow$ Photo | $0.8204 \pm 0.0202$ | $0.4868 \pm 0.0875$ | 0.3335 | 0.9615 |
| | Sketch $\rightarrow$ Sketch | $0.1066 \pm 0.0094$ | $0.0989 \pm 0.0125$ | 0.0076 | 0.0601 |

Table 2: **Calibration error vs. deviation from GDE under distribution shift:** Test error, disagreement rate, the gap between the two, and CACE for ResNet18 on PACS over 10 models each.

|  | **Test Error** | **Disagreement** | **Gap** | **CACE**[(100)] | **ECE** |
|---|---|---|---|---|---|
| AllDiff | $0.679 \pm 0.0098$ | $0.6947 \pm 0.0076$ | 0.0157 | 0.1300 | 0.0469 |
| DiffData | $0.682 \pm 0.0110$ | $0.6976 \pm 0.0074$ | 0.0150 | 0.1354 | 0.0503 |
| DiffInit | $0.681 \pm 0.0100$ | $0.5945 \pm 0.0127$ | 0.0865 | 0.3816 | 0.1400 |
| DiffOrder | $0.679 \pm 0.0097$ | $0.5880 \pm 0.0103$ | 0.0910 | 0.3926 | 0.1449 |

Table 3: **Calibration error vs. deviation from GDE for CIFAR100:** Test error, disagreement rate, the gap between the two, and ECE and CACE for ResNet18 on CIFAR100 with 10k training examples computed over 100 models.

## D.4    ERROR DISTRIBUTION OF ENSEMBLES

Here (Fig 11) we show the error distribution of the ensemble similar to N&B'20. The x-axis of these plots represent $1 - \tilde{h}_y(X)$ in the context of our work. As N&B'20 note, these plots are *not* bimodally distributed on zero error and random-classification-level error (of $\frac{K-1}{K}$ where $K$ is the number of classes). This disproves the easy-hard hypothesis as discussed in the main paper. As a side note, we observe that all these error distributions can be fit well by a Beta distribution.

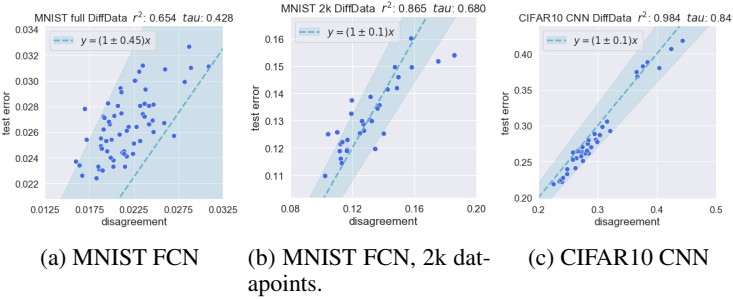

(a) MNIST FCN  (b) MNIST FCN, 2k datapoints.  (c) CIFAR10 CNN

Figure 10: Scatter plots for fully-connected and convolutional networks on MNIST and CIFAR-10 respectively.

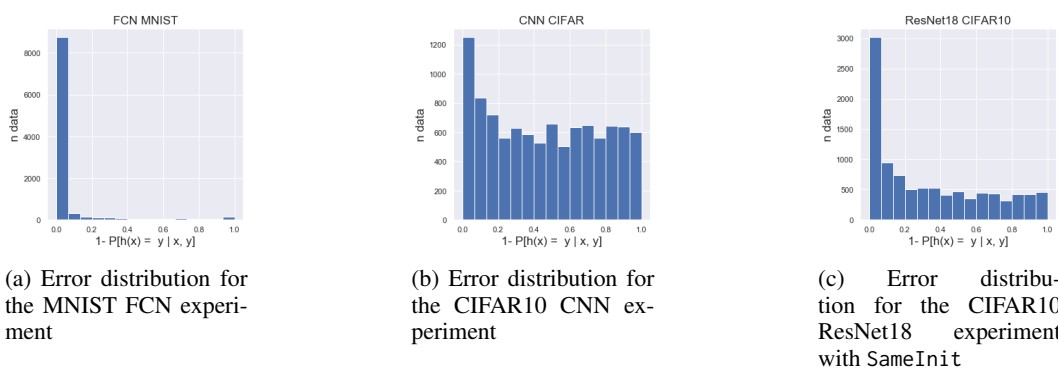

(a) Error distribution for the MNIST FCN experiment

(b) Error distribution for the CIFAR10 CNN experiment

(c) Error distribution for the CIFAR10 ResNet18 experiment with SameInit

Figure 11: Error distributions for different experiments

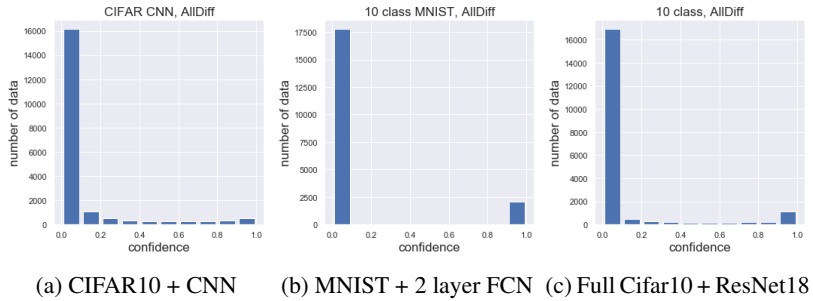

(a) CIFAR10 + CNN  (b) MNIST + 2 layer FCN  (c) Full Cifar10 + ResNet18

Figure 12: Histogram of calibration confidence for different settings.

### D.5 CALIBRATION CONFIDENCE HISTOGRAM

Here (Fig 12) we report the number of points that fall into each bin in calibration plots. In other words, for each value of $p$, we report the number of times the ensemble $\tilde{h}$ satisfies $\tilde{h}_k(x) \approx p$ for some $k$ and some $x$.

### D.6 COMBINING STOCHASTICITY

In Fig 13, for the sake of completeness, we consider a setting where both the random initialization and the data ordering varies between two runs. We call this setting the SameData setting. We observe that this setting behaves similar to DiffData and DiffInit.

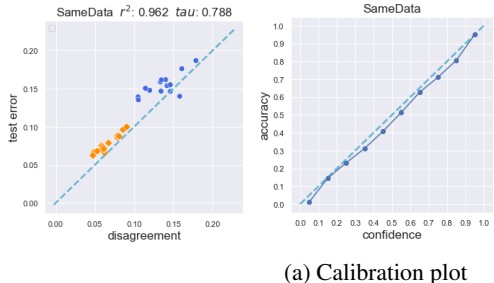

(a) Calibration plot

Figure 13: The scatter plot and calibration plot for model pairs that use the different initialization and different data ordering.

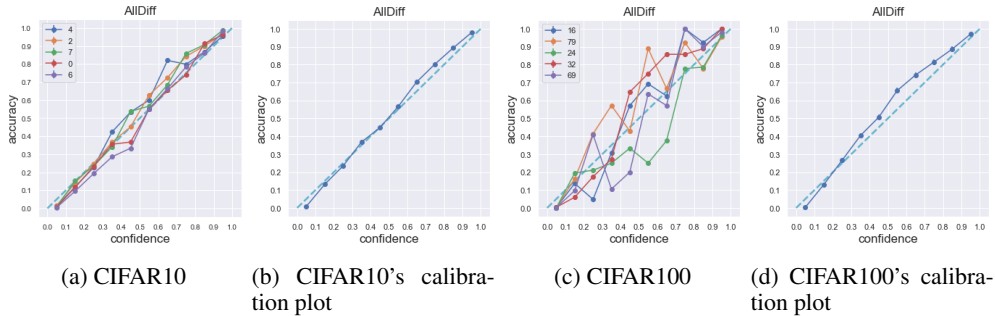

| (a) CIFAR10 | (b) CIFAR10's calibra-tion plot | (c) CIFAR100 | (d) CIFAR100's calibra-tion plot |

Figure 14: The calibration plot for 5 randomly selected individual classes vs the aggregated calibration plot for ResNet18 trained on CIFAR10 and CIFAR100.

## D.7 CLASS-WISE CALIBRATION VS CLASS-AGGREGATED CALIBRATION

In Fig 14, we report the calibration plots for a few random classes in the CIFAR10 and CIFAR100 setup and compare it with the class-aggregated calibration plots. We observe that the class-wise plots have a lot more variance, indicating that calibration within each class may not always be perfect. However, when aggregating across classes, calibration becomes much more well-behaved. This suggests that the calibration is smoothed over all the classes. It is worth noting that a similar effect also happens for ECE, although not reported here.

## D.8 CORRELATION VALUES IN FIG 1

In the main paper figures, we quantified correlated via $R^2$ and $\tau$ for scatter plots that include both data-augmented models and non-data-augmented models. Here, we report these values specific to each group of models.

|         | AllDiff | DiffData | DiffOrder | DiffInit |
|---------|---------|----------|-----------|----------|
| w/o aug | 0.888   | 0.977    | 0.728     | 0.923    |
| w aug   | 0.984   | 0.963    | 0.737     | 0.881    |
| both    | 0.986   | 0.998    | 0.941     | 0.983    |

Table 4: $R^2$ values

|         | AllDiff | DiffData | DiffOrder | DiffInit |
|---------|---------|----------|-----------|----------|
| w/o aug | 0.752   | 0.891    | 0.582     | 0.771    |
| w aug   | 0.829   | 0.891    | 0.626     | 0.650    |
| both    | 0.899   | 0.948    | 0.807     | 0.858    |

Table 5: $\tau$ values

## D.9 MORE THAN ONE PAIR OF MODELS

Here, we show the ResNet18 CIFAR-10 `DiffInit` experiments (no data augmentation) with only one pair of models v.s. the average of 4 pairs of models. We see that while 4 pairs of models does improve the correlation, the improvement is only marginal. This suggests that only using a single pair of models may be sufficient for estimating the generalization error.

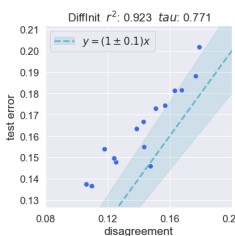
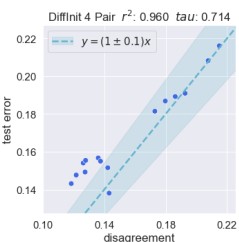

(a) The scatter plot for the disagreements of 1 pair of models.

(b) The scatter plot for the average disagreements of 4 pairs of models.

However, if we instead measure the distance from the $y = x$ line. Specifically, each pair's deviation is measured as

$$\frac{|\texttt{TestError} - \texttt{Disagreement}|}{0.5(\texttt{TestError} + \texttt{Disagreement})}. \tag{63}$$

The denominator normalizes the deviation so hyperparameters with different performance contribute equally. Then, we observed that 1-pair achieves an average deviation of $0.112$ while the 4-pair achieves $0.034$. This shows that if one is interested in estimating the exact generalization error, using more pairs of models would help.

