# OpenReview forum: "Assessing Generalization of SGD via Disagreement"
_ICLR.cc/2022/Conference — ICLR 2022 Spotlight_

### Official Review · Reviewer_APKH · 2021-11-01

**Correctness:** 4
**Technical Novelty And Significance:** 3
**Empirical Novelty And Significance:** 3
**Recommendation:** 8
**Confidence:** 3

**Main Review:**

Strengths:

I think the GDE phenomenon presented in this paper is quite novel, and in my opinion, this message could be quite interesting to the DL theory / phenomena community. The core theoretical observation that calibration implies GDE does have a simple proof, admittedly, but I think the contribution is more in identifying the correct design choices to allow this theorem (e.g. consider ensemble of one-hot predictions rather than ensemble of probabilities), which I think is somewhat non-trivial. Also, although it is still a bit opaque why calibration happens, the connection itself (between calibration and GDE) is kind of convincing and may motivate further research from both perspectives.

I also think the experiments in this paper are thorough and quite well-executed. The common variants (architectures, datasets, various notions of stochasticity, out-of-distribution performance) are sufficiently tested. Some of the more advanced questions arising from the theory (e.g. does ${\rm Dis}(h, h’)$ concentrate around its expectation, which I did wonder) are also tested via experiments.

Weaknesses:

It is a slight concern that the core observation in this paper is quite similar to Nakkiran and Bansal (2020)---Even though the authors already have a somewhat detailed comparison in the related work section, I am still left with some questions. Specifically, I get an idea of what is the difference in the empirical results (DiffData vs. the newly proposed DiffInit and DiffOrder), but am still unclear in the theoretical / formulational aspects: Did they propose GDE as-is (in expectation over h, h’), just restricted to the “DiffData” version? Also in the “they proved GDE specific to 1-NN models trained on different data”, did they not assume calibration but rather rely on something else to get that theorem?

Also, the main theoretical result is not end-to-end---calibration implies GDE, but no clues on when calibration happens in the first place. The authors discussed this on Page 9 and I suggest they also make sure various earlier claims reflect this limitation, when appropriate.

---
Update: I thank the authors for their response and updates to the paper. I am satisfied with the responses and would be glad to keep my current evaluation.

**Summary Of The Paper:**

This paper identifies and studies a new phenomena in the generalization of deep learning. First, it presents the observation of Generalization-Disagreement Equality (GDE) between same models trained with different stochasticity, strengthening the similar observation of Nakkiran and Bansal (2020). Second, it shows theoretically that calibration of ensembles (in both the per-class sense and a suitable aggregated sense) implies GDE, with accompanying experiments on their relationship.

**Summary Of The Review:**

This paper presents GDE, an empirical phenomena about the generalization of deep learning, with novel theoretical justifications via the connection to calibration, and thorough experiments.

---

> ### Author Response · Authors · 2021-11-12
> **Thank you for appreciating our work! More on the differences between ours and Nakkiran and Bansal'20's finding.**
>
> We are grateful for your encouraging feedback. We completely agree with your characterization of what aspects of our contribution are non-trivial.
>
> It’s of course very important that we clarify our differences from N&B’20 since we build on them, so we’re glad you raised some important questions here:
>
> \
> ***Question 1***
>
> > Did they propose GDE as-is (in expectation over h, h’), just restricted to the `DiffData` version?
>
> N&B’ 20 do formalize GDE in terms of the expectation for `AllDiff`. We formalize this just a bit more abstractly to include any stochasticity. **We did not mean to claim any sort of novelty in the math formulation -- we have rephrased the first para in Sec 4 to make this clearer.** Thanks for bringing this up.
>
>
> \
> ***Question 2***
>
> > “they proved GDE specific to 1-NN models trained on different data”, did they not assume calibration but rather rely on something else to get that theorem
>
> **N&B’s 1-NN proof does not make any explicit connections to calibration. They make an assumption about the input distribution that could further imply something philosophically like calibration**. Their assumption is about how the nearest neighbors are distributed: for a random point $X$, and datasets $S_1$ and $S_2$, the following two distributions must be nearly identical:
>
>  $(X, NN_{S_1}(X)) \approx (NN_{S_2}(X), NN_{S_1}(X))$  ----> (1).
>
> This assumption about the data points, then implies an assumption about the similarity in distribution of ground truth labels and predictions:
>
> $(Y, \hat{Y}_1) \approx  (\hat{Y}_2, \hat{Y}_1)$ ----> (2)
>  [this then gives way to GDE].
>
> Recall that calibration too, requires that the ground truth labels and the predicted labels are similar in distribution, but conditioned over the confidence level sets. i.e., we want
>
> $(Y | \mathbb{E}[\hat{Y}_1] = q)  \approx (\hat{Y}_2 | \mathbb{E}[\hat{Y}_1] = q)$ → (3)
>
> One could say that our calibration assumption in (3) in similar in spirit to their **implied** assumption in (2). In other words,
> - they don't make explicit connections to calibration
> - it seems reasonable to say that their main assumption (1) about the (input, nearest neighbor) distribution is philosophically stronger than our calibration assumption.,
>
> (As an aside, we have proven in our work that calibration is not a necessary condition for GDE.)
>
> \
> ***Question 3***
>
>
> > “no clues on when calibration happens in the first place”
>
> We agree this is a limitation of our work. While we admit it explicitly in the paper, we thoroughly agree that it’s important to be more upfront about this in the initial parts of the paper. As per your suggestion, **we have further highlighted this limitation right before the contributions and also in the third line of the contributions**. That being said, research on calibration and uncertainty estimation are rapidly growing within the deep learning literature and we hope our work can invite more attention to solving this important question.

---

### Official Review · Reviewer_DeNB · 2021-11-02

**Correctness:** 4
**Technical Novelty And Significance:** 3
**Empirical Novelty And Significance:** 3
**Recommendation:** 8
**Confidence:** 4

**Main Review:**

## Strengths
The main empirical observation, that the disagreement rate is nearly equal to test error across variations in architecture and hyperparameters, is surprising to me. (I would probably have expected disagreement to be more than 1.3x lower than test error for variations in random seed.) This experimental finding is a valuable contribution to the literature in and of itself. It is, as the authors note, a tantalizing connection between generalization and calibration. And the authors' explanation of this observation by reduction to a calibration property is clever and convincing (though as they note it only explains it in expectation). The paper is well-written.

The notion of "class-aggregated calibration" introduced in this paper is natural. The proofs that I checked appear to be correct. I was glad to see an attempt at an analysis of the variance of disagreement in the form of Corollary A.1.1, weak though it is.

## Weaknesses

The correlation between disagreement rate and test error is strong for random initialization and order but not as strong as it is for freshly drawn training sets. It also appears that much of the correlation might be coming from the affect of data augmentation being turned on or off. I would have liked to see plots where the disagreement rate is averaged across more than one pair of runs—how much stronger does the correlation with test error get? This is crucial to judging how effective disagreement would be in practice as a tool for predicting test error with unlabeled data. (The fact that there is correlation isn't surprising—it is intuitive that more error-prone classifiers will disagree more). More generally I would say that the main weakness of the paper is that it doesn't really attempt to present a strong case for the usefulness of the main observation in practice. That being said, the novelty of their empirical result and conceptual explanation are enough to make the paper worthy of acceptance in my opinion.

The proofs in the paper are quite straightforward (though somewhat long-winded because each small step is written explicitly); the theoretical innovation is in the statements, not the proofs.

## Other thoughts

As the authors mention, the study of calibration of neural network ensembles has primarily focused on their softmax probability outputs. Does softmax calibration imply anything about metrics of disagreement of softmax outputs (via, say, KL divergence)? I don't have much familiarity with the literature on calibration of neural network ensembles, so I don't know how novel this paper's discussion of that topic is.

**Summary Of The Paper:**

The authors build on a striking empirical observation of Nakkiran & Bansal—namely, that if two neural networks with the same architecture are trained on independently drawn training sets, achieving generalization error $\epsilon$, then their rate of disagreement on a test set is typically nearly equal to the test error of the two networks. In this paper, it is shown that this observation also holds even if the two networks are trained on the same data, but with different random seeds (data ordering and/or initialization).

The authors prove that test error will equal disagreement rate _in expectation_ for any stochastic learning algorithm whenever the algorithm's predictions have a certain calibration property.

**Summary Of The Review:**

I am recommending acceptance because this paper presents a surprising, simple experimental result and provides a satisfying partial explanation which connects test error disagreement to calibration. Ideally there would be more thorough experiments and more exploration of whether the experimental result is useful, but the paper is substantial enough that it is okay if such additions are left to future work. (Because of these qualms I would give the paper a score of 7 instead of 8 if it were possible to do so.)

---

> ### Author Response · Authors · 2021-11-12
> **Other concerns/questions**
>
> **Concern 2:**
> > correlation ... is strong for random initialization and order but not as strong as it is for freshly drawn training sets
>
> We acknowledge that our correlation is indeed weaker than the fresh-data-based estimate. We admit this a few times in the paper, including in the introduction when we say our plots ”slightly deviate“ from N&B’20. But perhaps we could’ve been clearer, so **we’ve added the following sentence in our introduction**: “Hence, ours is a more meaningful and practical estimator of test accuracy (albeit, a slightly less accurate estimate at that).”
>
> It’s worth noting though that, **it’s not surprising to see stronger correlation when using fresh labeled data**. A fresh labeled set has enough information to directly and trivially compute test accuracy (like a held-out estimate!). Our **disagreement estimate uses strictly less information about test data** (by using only unlabeled data) and is able to achieve reasonably similar correlation.
>
> **Concern 3:**
> > Does much of the correlation come from data augmentation being turned on/off?
>
> Great question.
> - We agree that data augmentation is a hyperparamter that does contribute in part to higher tau/r^2 values.
> - If we controlled for this hyperparameter and focused only on the other hyperparameters, the r^2 goes down but is still as large as [0.7.0.9], and tau ranges between [0.6, 0.77]-- please see Appendix C.3.
> - We believe this is still noticeably high enough to imply strong correlation.
> - Note that we also already had Fig 10 where for CNN we do not vary data augmentation and we observe very high values of 0.98 and 0.84.
> - Having said that, we understand it’s always possible for one hyperparameter to dominate the correlation, which we expect more so at such minute variations. Disentangling the effects of each hyperparameter requires much larger scale experiments beyond the scope of our part-theoretical, part-empirical work. The qualitative point we were hoping to convey is how we see reasonably strong correlation even for minute variations.
>
> Nevertheless, **to be transparent about this aspect, we have updated the main paper** under page 5 “observations”.
>
>
> \
>  **Concern 4:**
> > The fact that there is correlation isn't surprising—it is intuitive that more error-prone classifiers will disagree more
>
>
> We agree a correlation is not altogether _that_ surprising. But we'd like to mention a few minor points regarding why we think there's still a noteworthy surprise factor:
> - _A priori_, we’d expect the scatter plot to be **arbitrarily** spread between x=0 line and the x=2y line.  So we think there is still some surprise in correlation.
> - Nevertheless, we believe there's also an additional surprise in a **linear** correlation, and a further surprise in the proportionality constant lying in $1 \pm 0.3$.
> - We also expected that two networks trained on the same data would have very low disagreement compared to `DiffData`, and as a result higher variance in the disagreement values and poorer correlation.
>
> \
>  **Concern 5:**
> >“plots where the disagreement rate is averaged across more than one pair of runs—how much stronger does the correlation with test error get?”
>
>
>   The variance of the disagreement is small as observed in Table 1 where we consider a 100-member ensemble. Likewise, we have similar results for the scatter plot of distribution shift experiments. As per your request, **we've added plots in Appendix C.9**, specifically for `DiffInit` without data augmentation on ResNet18 + cifar10. **We observe only minor changes** after averaging over multiple (4) runs. We can't read much into these differences.
>
> Overall, we believe adding more pairs would not significantly improve the correlation. While writing the paper, we viewed this as an advantage since this suggests that we can get away with just two models for a reasonable estimate of generalization.
>
> But we understand your point of view as well. We are optimistic that future work could find a different workaround to bring the plot closer to the “y=x” line and rank the models better.
>
> *UPDATE*: Note that $R^2$ and $\tau$ only measure correlation, but doesn’t evaluate how close the points are w.r.t $x=y$ line. We evaluated “distance to x=y” through a different metric in C.9 and we observed that averaging over multiple runs does help (by up to a factor of 3). From a pure model selection perspective, $R^2$ and $\tau$ are sufficient, but this new metric demonstrates that using more pairs does benefit accuracy estimation.
>
>
> \
>  **Question:**
> > Does softmax calibration imply anything about metrics of disagreement of softmax outputs (via, say, KL divergence)?
>
> Our proof is general enough to conclude that for softmax calibration, the expected disagreement between softmax predictions is equal to the expected softmax test error.

---

> ### Author Response · Authors · 2021-11-12
> **Thanks for your detailed & positive feedback! Our method is a practical & simple approach to *blackbox* unsupervised accuracy estimation within-distribution!**
>
> We are glad that you find the observations and theoretical discussion valuable, and share our excitement about the implication of our findings!
>
> One of your biggest concerns is regarding practical implications. We agree we hadn't elaborated on this aspect in the paper initially, but we've updated the paper to address this.
>
> In short, we believe it’s valuable to look at our contribution as a solution to in-distribution “**unsupervised accuracy estimation**” (u.a.e) i.e., estimating test performance with only unlabeled test data.
>
> \
> **TL;DR:**
>
> U.a.e is **an existing practically well-motivated problem** (which we had mentioned briefly in the related work section). U.a.e is useful when we want **real-time evaluation, maintenance and debugging** (as motivated in [4]) and this does not require knowledge of test labels. We provide a very **simple** technique for in-distribution u.a.e, and ours works even with **blackbox** models where we don’t know weights/hidden representations. We think this makes our contribution practically relevant.
>
> \
> **Detailed answer:**
>
>  During test time, we may have access only to unlabeled data, because label annotation is costly (as motivated in [2]). Further, as [1,3] write, due to privacy considerations, we may only have access to blackbox models trained by a different organization, using training sets with proprietary labels, and with no access to hold-out sets from those organizations, even from the same distribution.  Similar motivations have been discussed in [5,6,7,8].
>
> In such a setting, how do we determine how well the blackbox models perform during deployment? How’d we pick the best blackbox model?
>
> Existing approaches to u.a.e apply only to specialized settings (e.g., language models with logical constraints) or require structural assumptions (understandably, because they often also care about distribution-shift) and/or require **specialized training** approaches. **While there is also a separate line of work like [9] that predicts generalization performance using neural network weights, they do not work for blackbox models**.
>
> Ours is a simple solution for u.a.e in the in-distribution setting: we simply measure disagreement between two **vanilla-trained** blackbox NN models. Overall, we hope that the
> - **simplicity** and
> - **the applicability to blackbox models**,
>
> **could make the case for its practicality, on top of the already-established practical motivation of unsupervised accuracy estimation**.
>
> **We have updated the introduction and the related work with a discussion of this -- thank you for bringing this up! We think this addition helps motivate our paper better and informs the reader as to why our experimental results could be useful in practice.** We are eager to know what you think.
>
> \
> **Reference**
>
> [1] Donmez, Lebanon, and Balasubramanian. Unsupervised supervised learning I: estimating classification and regression errors without labels. JMLR., 2010.
>
> [2] Chen, Liu, Avci, Wu, Liang, and Jha. Detecting errors and estimating accuracy on unlabeled data with self-training ensembles. arXiv preprint arXiv:2106.15728, 2021.
>
> [3] Jaffe, Nadler, and Kluger. Estimating the accuracies of multiple classifiers without labeled data. AISTATS 2015
>
> [4] ElSahar and Galle. To annotate or not? predicting performance drop under domain shift. EMNLP 2019
>
> [5]  Platanios, Poon,  Mitchell, and Horvitz. Estimating accuracy from unlabeled data: A probabilistic logic approach. NeurIPS 2017.
>
> [6] Steinhardt and Liang. Unsupervised risk estimation using only conditional independence structure. NeurIPS 2016.
>
>
> [7] Schelter, Rukat, and Bießmann. Learning to validate the predictions of black box classifiers on unseen data. ICMD 2020.
>
>
> [8]  Chuang, Torralba, and Jegelka. Estimating generalization under distribution shifts via domain-invariant representations. ICML 2020.
>
>
> [9] Jiang, Neyshabur, Mobahi, Krishnan, and Bengio. Fantastic generalization measures and where to find them. In ICLR 2020.

---

> > ### Comment · Reviewer_DeNB · 2021-12-01
> > **Clarification**
> >
> > Thank you for the helpful response, and for making the edits to your paper.
> >
> > I'm a little confused, actually, about the precise black box u.a.e. setting you're envisioning. Are you assuming the outside organization has trained the model twice using two random seeds, with their proprietary training set? (...in which case we need to trust that the two seeds are actually random.) Or that it's the evaluator's job to train the model twice? (...in which case the evaluator needs access to the proprietary training labels and to the model.)

---

> > > ### Author Response · Authors · 2021-12-01
> > > **Motivation for blackbox approach**
> > >
> > > Thanks a lot for reading our response and getting back to it.
> > >
> > > Apologies for the confusion!
> > >
> > > Privacy is one way to motivate why we are interested in black box models. In that particular case, we were envisioning the former, where we imagine having an agreed-upon contract with the org to release two independently trained black box models, but not the weights or the labeled training data themselves. (Hopefully, establishing this trust is not contradictory to the goal of privacy)
> > >
> > > Having said that, there are also other reasons to want a black box approach, such as usability by non-experts and computational efficiency. For example, Schelter et al., (see [7] above) point out the case where non-expert engineers use external services to train blackbox models. We simply require them to train twice with two random seeds. A blackbox approach may also be more efficient during test-time as we don't need to process the weights in anyway.
> > >
> > > Nevertheless, we agree that in the example motivating the privacy considerations, it's indeed a limitation that we've to establish trust in the outside org.
> > >
> > > But more broadly, we hope that our method is complementary to other u.a.e approaches. It has both advantages (e.g., simplicity, blackbox, no structural assumptions) and disadvantages (e.g., requires two independently trained models, in-distribution) which future work could improve upon.
> > >
> > > Please let us know if you have any further questions!

---

> > > > ### Comment · Reviewer_DeNB · 2021-12-01
> > > > **almost not confused**
> > > >
> > > > Ah, okay, then I guess what's tripping me up is that if you trust the org, you might as well just ask them to tell you the model's accuracy on on their hold-out set. Unless I'm still wrong about some aspect of the scenario? In any case, I agree about the advantages and disadvantages of your method, and I consider discovering and partially explaining the emergent phenomenon sufficient in and of itself for acceptance regardless of the exact level of practicality of applications.

---

> > > > > ### Author Response · Authors · 2021-12-01
> > > > > **When hold-out data exists...**
> > > > >
> > > > > Agreed. Broadly, if there was a hold-out set, in-distribution u.a.e would be moot from a practical point of view. However, we think that in cases where preserving a hold-out dataset is expensive and where one would rather use that to train the model, then our approach (and in-distribution u.a.e) would become practically relevant.
> > > > >
> > > > > Thanks again for engaging with us during this process, and for your honest feedback!

---

### Official Review · Reviewer_5dWv · 2021-11-02

**Correctness:** 4
**Technical Novelty And Significance:** 4
**Empirical Novelty And Significance:** 4
**Recommendation:** 8
**Confidence:** 4

**Main Review:**

 **Strength**

The paper marks a significant advance in the understanding of the generalization of deep neural networks trained with SGD. Specifically, a theoretical connection between generalization and calibration is provided. This opens up new opportunities and inspiring directions for the study of generalization in deep learning.  The theoretical analysis is sound and the experimental results are convincing. The paper is also beautifully written and a pleasure to read.

**Weakness**

As the authors also acknowledge, why and when SGD (or any given learning algorithm) is well-calibrated remains unclear and requires further characterization. It is also curious that the prediction disagreement of a pair of learned models from SGD appears to have low variance. Finally to what extent the development of this paper will truly impact further understanding of deep learning is yet to be seen. But rather than considering these as weaknesses, I tend to think of them more positively, as being inspiring and pointing to new directions. The contributions of this paper in my opinion well justify an acceptance decision.

The writing of the paper is occasionally sloppy, for example, in the proof of Theorem 4.1, the authors speak of ${\scr D_q}$ as if it is a set (rather than a distribution), in, for example,  "calculate the error on ${\scr D_q}$", "a $q$ fraction of ${\scr D_q}$" etc.

The authors are also encouraged to provide more intuition in the main body of the paper regarding the key insights in the proofs. Although the proofs are not difficult to follow, the reader is likely to be buried in their technical details without truly appreciating the fundamental reasons for which the theorems are true.

Finally, the developed theory is not limited to the context of SGD. It might be a good idea to reflect the generality of the theoretical results in the paper title. -- a suggestion, not a weakness.

**Overall Recommendation**

Overall I recommend the acceptance of this paper without any reservation.


**Summary Of The Paper:**

This paper builds upon and extends from Nakkiran/Bansal (2020). First, it shows empirically that models learned from two independent runs of SGD on the same training set have their prediction disagreement highly correlated (and nearly equal) to the test error of the models.  It then attempts a theoretical explanation for this phenomenon. Under two notions of "calibration", the paper proves that when the learning algorithm is well-calibrated, a "generalization disagreement equality" holds (i.e., disagreement in prediction is equal to the test error in expectation).  The paper performs further experiments and provides empirical evidence suggesting that SGD is well calibrated.  The paper also proves that the gap between the test error and the prediction disagreement is upper bounded by a calibration error.

**Summary Of The Review:**

The paper contains novel and inspiring empirical observations, which are justified by establishing a relationship between generalization and prediction disagreement of two models learned independently using SGD.  The paper is also well written.

---

> ### Author Response · Authors · 2021-11-12
> **Thank you for your thoughtful and encouraging feedback!**
>
> We are really happy to see that you view the weaknesses in these findings as opportunities! Regarding your other concerns, we've done our best to address them. Details below:
>
>
>
> - We sincerely apologize for the sloppy writing. **We have rewritten the proof sketch/notations to fix the issue you pointed out.**
>
>
>
> - Also, we wholeheartedly agree that our paper would benefit from some more discussion of the fundamental intuition behind the theory. **We have elaborated on the brief intuition that we originally had after Theorem 1’s proof sketch (1 para → 2 para) and added some extra intuition for class-aggregated calibration in the appendix (due to space constraints).** Thank you for the thoughtful suggestion. We believe this makes our paper more accessible.
>
>
>
> - Regarding the title, we are pleased to see your suggestion. Since our experiments are purely focused on SGD and deep learning, we are worried that our title might be an overclaim if we drop SGD from it. If you have an alternative suggestion or if you feel strongly about this, we’d love to hear that!

---

> > ### Comment · Reviewer_5dWv · 2021-12-03
> > **Thanks for the reply**
> >
> > Thanks for your effort in replying to reviews and revising the paper.  I confirm that I will keep my original score. Cheers.

---

### Official Review · Reviewer_sYb8 · 2021-11-03

**Correctness:** 4
**Technical Novelty And Significance:** 3
**Empirical Novelty And Significance:** 3
**Recommendation:** 8
**Confidence:** 3

**Main Review:**

Strengths:
- The connection between calibration and generalization is novel. The theoretical results are sound.
- This paper shows stronger results than prior work (Nakkiran & Bansal 2020) with more practical implications-- being able to approximate the test error without a separately labeled dataset, given calibration.
- Experiments are through and sound.
- Paper is well written and easy to follow.

Weaknesses:
The data for DiffOrder and DiffInit deviates from the y=x curve more than the other two cases where a fresh dataset is used, so in practice, it’s still more accurate to use a fresh-labeled dataset.

Comments:
- On the last line on page 6 “namely ½ if x is hard, and 0 if easy”, is ½ supposed to be 1/k where k is the number of classes?
- Page 7: typo in “even though an ensembe of one or…”

--- Update ---

I have read all the reviews and I am satisfied with the authors' responses.


**Summary Of The Paper:**

This paper shows empirically that the test error of a network is approximately equal to the disagreement rate between two separate training runs of the network, measured on unlabeled test data. This is true even when the two training runs are on the same training set, and with varying sources of stochasticity (e.g. random seed, ordering of training data). The authors show theoretically that this phenomenon is due to ensembles trained with SGD being well-calibrated.

**Summary Of The Review:**

This paper has novel contributions with both theoretical and practical implications. I recommend acceptance.

---

> ### Author Response · Authors · 2021-11-12
> **Thank you for your positive review! Regarding practicality:**
>
> Thanks a lot for your time and for your encouraging feedback!
>
> We'd like to respond to your thoughts below.
>
> > “it’s still more accurate to use a fresh-labeled dataset”:
>
> You’re right that our unlabeled-data-based estimate is indeed less accurate. **We believe though that our estimate is actually more practical** than the fresh-data-estimate because requiring labels on a **fresh held-out dataset is expensive in practice** [2] or even inaccessible due to privacy considerations [1, 3] as has been motivated in prior accuracy estimation work.
>
> A labeled test dataset has enough information to (trivially and directly) compute the test accuracy while the **unlabeled dataset is strictly less informative** and naturally requires a more clever approach to estimation. We think **the loss in estimation accuracy power is a tradeoff for more practicality**, though future works may still address this issue.
>
> Nevertheless, we completely agree and acknowledge the drop in estimation accuracy!
> This is something we too admit a few times in the paper, including in the introduction when we say our plots ”slightly deviate“ from N&B’20. But we could’ve been clearer, so **we’ve further added the following clause in our introduction**:
> > Hence, ours is a more meaningful and practical estimator of test accuracy (albeit a slightly less accurate estimate at that).
>
>
> ===
>
> Regarding other comments:
> - Great catch! It should be $\frac{K-1}{K}$ and we have corrected it in the paper. It was ½ because we were thinking about binary classification.
> - Fixed the typo, thanks!
>
> \
> **References** [also cited in the paper]:
>
> [1] Pinar Donmez, Guy Lebanon, and Krishnakumar Balasubramanian. Unsupervised supervised learning I: estimating classification and regression errors without labels. JMLR., 2010.
>
> [2] Jiefeng Chen, Frederick Liu, Besim Avci, Xi Wu, Yingyu Liang, and Somesh Jha. Detecting errors and estimating accuracy on unlabeled data with self-training ensembles. arXiv preprint arXiv:2106.15728, 2021.
>
> [3] Ariel Jaffe, Boaz Nadler, and Yuval Kluger. Estimating the accuracies of multiple classifiers without labeled data. AISTATS 2015

---

### Author Response · Authors · 2021-11-22
**Quick summary of changes**

We’d like to once again express our gratitude to the reviewers for their encouraging reviews and for their many constructive suggestions.

Below is a quick summary of how we’ve modified our paper to incorporate those suggestions:
1. We’ve added why our finding has practical relevance: it provides a simple blackbox approach to in-distribution unsupervised accuracy estimation.
2. We’ve rewritten the proof sketch and added more intuition in the main paper.
3. We have added an extra scatter plot in the appendix averaging over 4 pairs of models rather than just a single pair. We see that the plots get slightly closer to x=y line (although correlation-wise there’s not much difference). We also report correlation values individually for data-augmented and non-data-augmented models.
4. We’ve made it clearer, especially in the introduction that (a) our disagreement metric can be slightly less accurate than one that uses labeled test data (as is expected) and that (b) we don’t explain when/why calibration holds.
5. We've clarified our differences from Nakkiran and Bansal '20, especially the fact that they don't use calibration to prove their theorem for 1-nearest neigbhors.

---

### Public Comment · ~Andreas_Kirsch1 · 2022-02-07
**A Short Note on the Paper**

**Update 2022/11: Below paper has been published at TMLR now under https://openreview.net/forum?id=oRP8urZ8Fx**

Thank you very much for writing this paper. We've found it very interesting and spent a bit of time understanding the theory presented in it and the implications more deeply.

We have written up these thoughts in https://arxiv.org/abs/2202.01851. We hope this benefits other readers who want to understand this spotlight paper better.

**A Note on "Assessing Generalization of SGD via Disagreement"**:

---

We show that the approach suggested might be impractical because a deep ensemble's calibration deteriorates under distribution shift, which is exactly when the coupling of test error and disagreement would be of practical value.

We present both theoretical and experimental evidence, re-deriving the theoretical statements using a simple Bayesian perspective and show them to be straightforward and more generic: they apply to any discriminative model -- not only ensembles whose members output one-hot class predictions.

The proposed calibration metrics are also equivalent to two metrics introduced by Nixon et al. (2019): 'ACE' and 'SCE'.

---

Especially the first insight above should give pause. The proposed metrics deteriorate substantially as the disagreement rate (=predicted error) increases. The implied bounds thus widen as well and do not provide a guarantee that the disagreement rate/predicted error is a good predictor of the test error. Indeed, the gap doubles in our experiment on CINIC-10 with an ensemble of models trained on CIFAR-10.

I hope this is useful.

Best wishes,\
 Andreas

---

> ### Public Comment · ~Vaishnavh_Nagarajan1 · 2022-02-08
> **Thank you! (Part 2)**
>
>
> Other points
> ===
>
> **Triviality**: “[our notion of] calibration is so strong that the connection trivially follows”.
>
> - Indeed, it’s true that the relationship in hindsight seems quite straightforward.
> At the point of publication of our paper however, GDE was only empirically known – and that too only for independent-data-runs – but no one knew why it held. The role of calibration was far from obvious then.
> - Furthermore, this strong notion of calibration does hold in practice in-distribution,  thus shedding some insight on an empirical phenomenon. **If you agree, we sincerely hope that your note could also give credit to what is indeed non-trivial in our findings so that readers may gain a more balanced understanding of our work from your note**.
>
> **Proof simplification**: Thanks again for your neat simplification of our proof!  Perhaps it’s necessary for us to emphasize that we did our best to be clear about the simplicity of the math and not obfuscate it: we had noted that the central proof idea is fairly simple, and provided a proof sketch for a simple case so that readers do not think that there’s anything fancy going on. (We’d like to also add that the length of the proof in the appendix is more so because of our interest in laying out every step explicitly.)
>
> **Calibration assumption is circular as it requires labeled data**: We had noted in multiple places that to use GDE in practice, we need to know a priori that the ensemble is well-calibrated.
> - “While our estimate applies to the original training process, our guarantee applies only if we know a priori that the training procedure results in well-calibrated ensembles.” (in related work)
> -  “First, we do not provide a theoretical characterization of when we can expect good calibration (and hence, when we can expect GDE to hold).“ (right before sec 6)
>
> This is indeed a drawback in distribution-shift settings as it’d require labeled data to check for the condition. We will state this drawback more clearly in future versions of the paper.
>
>
> **Generality**: With regard to your conclusion “we also show that the theoretical results of Jiang et al. (2021) are more general and hold even for single deterministic models…” It seems to us that we may also be noting something similar, that our theory doesn’t apply only to ensembles:
>
> - In the remarks at the end of Sec 4.1 in our ICLR submission, “For example, if h˜ was an individual calibrated neural network whose predictions are given by softmax probabilities, then GDE holds for the neural network itself: the disagreement rate between two independently sampled one-hot predictions from that network would equal the test error of the softmax predictions”.
>
> **CACE vs SCE**: Thank you for bringing to our attention the implementation of SCE. The equivalence of their implementation and CACE was actually very non-obvious to us, as the original definition of SCE in their paper (which we mentioned in our ICLR appendix/arxiv main paper as the most closely related metric), _does_ differ from CACE.  But, as you mention they also develop a slightly more general _implementation_ of SCE that does appear to match CACE.  This is great to know, and thanks for pointing it out!
>
>
>
>
> **Concluding thoughts**:
> ===
>
> Again, we thank you for your criticism of our work, all of which we agree with! We do believe it’s worth noting that we had already gone to great lengths to highlight all the above limitations (and many others!) in multiple places (in the introduction, towards the end and also in the main text). We were also particular about specifying the limited nature of the contributions in the case of OOD data as we didn’t want to mislead the reader. **We’re afraid that the note as it is currently written inadvertently suggests that our paper did not lay out its limitations honestly and that it over-promises for OOD data, which is unfortunate.**
>
> We humbly invite you to reconsider whether it’s possible to present a more balanced view of our work and whether a call for a “pause” is really necessary for the way we presented our findings. Of course, we would love to know which parts of the paper, if any, can be edited to avoid this confusion for future readers. We again appreciate your significant interest in our paper, and we are grateful for the simplified proof. We are happy to address any further questions you have and sincerely hope you understand our concerns.
>
>
> Reference:
> Garg et al 2021. RATT: Leveraging Unlabeled Data to Guarantee Generalization
>
>
> P.S: We think you might find this concurrent paper (which we’ve cited) interesting as they present a disagreement-based technique on out-of-distribution data. https://arxiv.org/abs/2106.15728
>
>
> Regards,
> Authors

---

> > ### Public Comment · ~Andreas_Kirsch1 · 2022-04-25
> > **Reply**
> >
> > Dear Authors,
> >
> > First, apologies for the late reply. I came down with COVID, and the recovery and return to productivity took a bit—and then had to perpetually catch up on lots of things in the last weeks.
> >
> > Thank you very much for your detailed reply and for sharing additional literature!
> >
> > The latest ICLR version on OpenReview clarifies a lot of concerns. The note was based on the earlier arxiv version https://arxiv.org/abs/2106.13799 (v1) (to be citable).
> >
> > In general, however, there persists an issue of circularity:
> >
> > 1. The test error can be bound using the introduced calibration metrics.
> > 2. The calibration metrics are aggregated test errors and require labels themselves.
> >
> > There is no reason to not simply compute the test error directly when one has access to labels.
> >
> > Anything beyond that requires an inductive step that calibration stays constant under distribution shift. And only this case is really of interest to practitioners, I would posit. Yet, calibration decreases as the predicted error increases, as shown empirically in the note in one first example. This is not helpful, and from reading the paper one would still assume that this would not be the case.
> >
> > Second, the disagreement rate actually refers to the predicted error. The statements in the paper are all about the predicted error. Only because the ensemble members are one-hot, does the predicted error equal the disagreement rate.
> >
> > Finally, the challenge I see because of the simplified proof is that it is too simple: the definitions do all of the “heavy-lifting”. Class-wise/class-aggregated calibration correspond to GDE.
> >
> > As a side-note, that Deep Ensembles are well-calibrated as a well-known fact (citing Lakshminarayanan et al., 2017) only refers to ECE and not the stronger calibration metrics (class-wise/class-aggregated calibration) proposed in the paper.
> >
> > From reading the latest version, subjectively, I still get the impression that one should be able to trust the predicted error to predict the test error, so I believe the call for a pause and reflection in the note remains valid, at least as one data point.
> >
> > Thanks so much, and best wishes,\
> >  Andreas

---

> ### Public Comment · ~Vaishnavh_Nagarajan1 · 2022-02-08
> **Thank you (Part 1)**
>
> Thanks a lot for your interest in our work. We are really happy to see your simplified proof!
>
> Although we’re naturally a bit biased in our viewpoint, our reading of your note suggests to us that you interpreted our paper as saying several things we absolutely wouldn’t have wanted to imply (and indeed, which we explicitly tried to avoid).  This includes, e.g., your statement that our theorem intends to imply anything about OOD performance based on in-distribution calibration.  We’ll cover this a lot more below.   It’d be great to clear up these points.
>
> But overall, we don’t think we really disagree on many factual issues here (we in fact whole-heartedly agree with many of the elements you point out, including the fact that _the actual implementation_ of SCE equals CACE!).  **It’s just that we had tried to _explicitly_ point out many of the shortcomings that you mentioned.**  Hopefully we can clear up any misunderstandings here.  Please do let us know which portions of our paper implied these things to you, as they are absolutely not the intention of the paper.
>
>
>
> **Regarding our main claim restated** in your note as “calibration error for in-distribution data bounds the absolute difference between the test error and ‘disagreement rate’ across datasets with increasing distribution shift—independent of the distribution shift”
>
> If we understand correctly, the restatement is “GDE holds on any test distribution as long as calibration holds in-distribution”. **We’d like to point out that our statement is different**: “if calibration holds on a particular distribution, then GDE holds on **_that_** distribution”. This if-then statement holds for any distribution (this is what we meant by we don’t need any assumptions about the distribution). **We have definitely not claimed that GDE will hold under distribution shifts unconditionally** — we agree with you that this would of course be incorrect. If you believe there’s any text in our paper that suggests otherwise, please do let us know and we will be more than glad to fix it.
>
> **Regarding our main focus and the role of OOD experiments**:
> As stated in our introduction, we wanted to uncover and understand a surprising empirical in-distribution phenomenon. This also happens to be of use in the in-distribution setting. We must emphasize that *the use-case is in a similar vein as Garg et al., 2021 where they develop an in-distribution predictor of accuracy using unlabeled data*. Given that most training-data-based predictors for in-distribution generalization seem to struggle at predicting generalization, Garg et al., 2021 marks a step towards trying and using unlabeled data to do the same, which *is still a meaningful thing to do*. Our paper explores a different approach to this idea.
>
> While we discussed the practical use-case of our finding, we focused mainly on in-distribution, but we also mentioned the OOD setting but only as an aside. We believe we did our best to be clear and precise about the preliminary nature of our OOD observations, and to not over-claim (but we will still make it clearer). Here are the only references to the OOD results in the introduction:
>
> - “Furthermore, we show that under certain training conditions, these properties even hold on many kinds of out-of-distribution data **in the PACS dataset (Li et al., 2017), albeit not on all kinds**.”
> - At the end of the intro: “We present **preliminary observations** showing that GDE is approximately satisfied even for **_certain_** distribution shifts within the PACS (Li et al., 2017) dataset. This implies that the disagreement rate can be a promising estimator even for out-of-distribution accuracy.”
>
> We believe we’ve also been particularly careful about the in-distribution nature of our results even in our rebuttals.
>
> **We would totally agree that a call for “a pause” would be appropriate had our theorem been incorrect or had we made unjustified claims on distribution shifts.** We’d like to note that to the best of our knowledge, neither of this seems to be the case. Nevertheless, since it seems to have caused confusion, we completely understand there’s room to improve the clarity here.

---

### Decision · Program_Chairs · 2022-01-20

**Decision:**

Accept (Spotlight)

**Comment:**

This article introduces an interesting variant of the work of Nakkiran & Bansal (2020). It shows empirically that the test error of deep models can be approximated from the disagreement on the unlabelled test data between two different trainings on the same data. The authors then show theoretically that a calibration property can explain such behaviour, and they report experiments showing that the relationship does exist in practical situations.  All reviewers agree on the practical and theoretical value of the article, which is very well organised and written. The ideas developed here are likely to lead to further work in the future, and they clearly deserve to be published at ICLR.

I agree with one of the reviewers that the title is somewhat misleading, as the reader expects an analysis based on SGD. The title could be shortened to "Assessing Generalization via Disagreement", and the experimental restriction to SGD could be mentioned in the abstract.